# Environmental modulation of global epistasis in a drug resistance fitness landscape

Juan Diaz-Colunga [1,2,4] ✉, Alvaro Sanchez [2,4] ✉ &
C. Brandon Ogbunugafor [1,3] ✉

Interactions between mutations (*epistasis*) can add substantial complexity to genotype-phenotype maps, hampering our ability to predict evolution. Yet, recent studies have shown that the fitness effect of a mutation can often be predicted from the fitness of its genetic background using simple, linear relationships. This phenomenon, termed *global epistasis*, has been leveraged to reconstruct fitness landscapes and infer adaptive trajectories in a wide variety of contexts. However, little attention has been paid to how patterns of global epistasis may be affected by environmental variation, despite this variation frequently being a major driver of evolution. This is particularly relevant for the evolution of drug resistance, where antimicrobial drugs may change the environment faced by pathogens and shape their adaptive trajectories in ways that can be difficult to predict. By analyzing a fitness landscape of four mutations in a gene encoding an essential enzyme of *P. falciparum* (a parasite cause of malaria), here we show that patterns of global epistasis can be strongly modulated by the concentration of a drug in the environment. Expanding on previous theoretical results, we demonstrate that this modulation can be quantitatively explained by how specific gene-by-gene interactions are modified by drug dose. Importantly, our results highlight the need to incorporate potential environmental variation into the global epistasis framework in order to predict adaptation in dynamic environments.

The topography of genotype-phenotype maps has critical consequences for the predictability of evolutionary trajectories. This topography emerges from complex interactions between genetic elements, from single nucleotides to protein residues and metabolic pathways[1–5]. Despite this complexity, recent work has shown that epistasis−the nonlinear interaction between parcels of genetic information−often has a "global" component[6–15], emerging in the form of simple relationships between the fitness effect of a mutation and the fitness of the genetic background where it arises (Fig. 1a–c). Global

epistasis has become a central concept in modern conversations surrounding the fundamentals of evolutionary processes. Negatively sloped correlations have been more commonly reported, both in the form of *diminishing returns* and *increasing costs* epistasis (where the fitness effect of a mutation becomes less beneficial or more deleterious, respectively, in fitter genetic backgrounds)[6–8, 11–13]. Positive slopes (*increasing returns* or *decreasing costs* epistasis) have also been found in low-dimensional landscapes[7,16,17], and theory suggests they may be more common near fitness peaks[5,18,19].

[1]Department of Ecology & Evolutionary Biology, Yale University, New Haven, CT 06511, USA. [2]Department of Microbial Biotechnology, Spanish National Center for Biotechnology CNB-CSIC, 28049 Madrid, Spain. [3]Santa Fe Institute, Santa Fe, NM 87501, USA. [4]Present address: Institute of Functional Biology and Genomics IBFG-CSIC, University of Salamanca, 37007 Salamanca, Spain. ✉e-mail: juan.diazcolunga@yale.edu; alvaro.sanchez@usal.es; brandon.ogbunu@yale.edu

**Fig. 1 | Mutations exhibit variation in the strength of global epistasis. a** We reanalyzed a dataset consisting of 15 genotypes of the *P. falciparum* parasite, each carrying a different combination of four mutations: C59R, I164L, N51I and S108N. **b** We examined the fitness effect of a focal mutation (in this illustration, mutation C59R) in different genetic backgrounds carrying combinations of the other three mutations. Here, colored/gray loci represent the presence/absence of the mutation. **c** In the absence of pyrimethamine, a negative correlation is observed between the fitness effect of mutation C59R and the fitness of its genetic background. We quantify the strength of epistasis as the variance in the mutation's fitness effects relative to the variance in fitness across its genetic backgrounds (var $\Delta f$ / var $f(B)$). The degree to which epistasis is global is quantified as the $R^2$ of the linear regression between the fitness effect of the mutation and the fitness of its genetic background. **d** The variance ratio and the $R^2$ can be seen as a mutation's "coordinates" in a conceptual "epistasis map".

The emergence of global epistasis evidences the existence of regularities in genotype-phenotype maps[6–10], which have been leveraged in recent methodologies to infer complete adaptive landscapes under defined, steady environmental conditions[20–22]. Yet, the magnitude of mutational fitness effects[23–26] and epistatic interactions[27–32], and thus the topography of fitness landscapes[33–35], can generally depend on environmental variables. Populations are often subject to natural or anthropogenic environmental fluctuations which can dictate their evolutionary fate[26, 29,36,37]. Importantly, this effect has been widely described in the evolution of antimicrobial resistance (AMR)[29,32,38–40]. Learning how global epistasis patterns may be shaped by environmental factors is therefore critical to our ability to predict adaptation in changing environments, and especially in the context of AMR evolution. This is yet an open question largely because understanding how global epistasis emerges from fine-grained genetic interactions (which might be subject to environmental regulation) is still in its early days.

To address these questions, here we analyze a previously published fitness landscape[31,34,41] consisting of 15 genotypes of the *P. falciparum* malaria parasite. Each genotype carries a different combination of four mutations (amino acid substitutions C59R, I164L, N51I and S108N) at four sites of the *P. falciparum* gene for the dihydrofolate reductase (DHFR) enzyme, which confers resistance to several antifolate drugs (Fig. 1a). These particular four mutations have been associated to antimalarial drug resistance in numerous studies across multiple geographical locations[42–45]. The 15 genotypes were cultured in a concentration gradient of pyrimethamine or cycloguanil (antifolate drugs commonly used to treat malaria), and fitness was quantified as the growth rate relative to that of the slowest growing genotype in the absence of drug. Notably, epistasis makes it so the genotype carrying all four mutations has lower fitness than the sum of all mutations' individual fitness effects at the highest drug concentrations (which can be explained by resistance mutations often being redundant[46,47]); but also reduces the deleterious effect of the four mutations when there is no drug in the environment (Supp. Figure 1). This observation highlights the need to characterize how drug dose modulates epistasis in this particular landscape in order to understand the emergence and prevalence of resistant *P. falciparum* genotypes.

We thus asked whether the concentration of drug in the environment may modulate the patterns of global epistasis observed for a particular mutation and, if so, whether we can trace back the origins of this modulation to specific gene-by-environment interactions. For each of the four mutations, we focused on (a) to what extent the fitness effect of the mutation depends on its genetic background (i.e., how much epistasis there is), (b) how correlated the fitness effect of the mutation is with the fitness of its genetic background (i.e., to what degree epistasis is global), and (c) what the shape of global epistasis is (e.g., diminishing returns, accelerating returns, etc.).

Our results show that drug concentration strongly modulates global epistasis in this particular landscape. Extending previous theoretical results, we demonstrate that this modulation can be explained by specific gene-by-gene interactions and how they are affected by the

environment. In particular, we mathematically define a set of "effective" genetic interactions, showing that their distribution determines the strength and shape of global epistasis across drug doses. Our results highlight an avenue for quantitatively connecting fine-grained gene-by-environment and gene-by-gene-by-environment interactions with the emergence of global epistasis and the topography of fitness landscapes.

## Results

### Mutations exhibit variation in the strength and shape of global epistasis

We first asked to what extent mutations in our landscape exhibit variable fitness effects depending on their genetic background, that is, what the magnitude of epistasis is in this system. For a given mutation (which we denote as the focal mutation $i$), we refer to the set of genotypes not carrying that mutation as the *genetic backgrounds* (Fig. 1b). We denote $f(B)$ the fitness of one of such backgrounds $B$. Calling $B+i$ the genotype resulting from adding mutation $i$ to the genetic background $B$, the fitness effect of mutation $i$ is quantified simply as: $\Delta f_i = f(B+i)-f(B)$, that is, the difference in fitness between genotypes $B+i$ and $B$ (see Fig. 1b for an illustration with C59R as the focal mutation). If a mutation had only weak epistatic interactions with other loci, we would see that its fitness effect $\Delta f_i$ remains roughly constant across all genetic backgrounds (i.e., the effect of the mutation is essentially additive). On the other hand, high levels of epistasis would make this fitness effect largely dependent on the presence or absence of other mutations, exhibiting large variation across backgrounds. We thus quantified the strength of epistasis for a focal mutation $i$ as the variance of $\Delta f_i$ relative to the variance in fitness across the genetic backgrounds of that mutation: $\text{var}\,\Delta f_i/\text{var}\,f(B)$ (see Fig. 1c for the case with C59R as the focal mutation).

We also asked the extent to which the fitness effect of a mutation may be well predicted by a simple linear model linking it to the fitness of its genetic background, that is, to what degree is epistasis global. Note that, in general, the relationship between $\Delta f$ and $f(B)$ need not always be truly linear, and other studies have found different types of

relationships between the two magnitudes[14,15]. The linear fits to our data should be interpreted simply as the most parsimonious statistical models one can use to link the fitness effect of a mutation to the background fitness.

If epistasis was strong for a given mutation (large $\text{var}\,\Delta f_i/\text{var}\,f(B)$) and we observed a high correlation between $f(B)$ and $\Delta f$, we would interpret this as epistasis being largely global: while the fitness effect of the mutation strongly depends on the genetic structure of its background, it can still be well estimated from the background fitness through a simple linear model (Fig. 1c). If, on the other hand, the correlation between $f(B)$ and $\Delta f$ was weak, epistasis would be largely idiosyncratic (as opposed to global). We quantified the degree to which epistasis is global for a given mutation as the coefficient of determination ($R^2$) of the regression between $f(B)$ and $\Delta f$ for that mutation (Fig. 1c).

We propose that these two quantities (the variance ratio $\text{var}\,\Delta f_i/\text{var}\,f(B)$ and the $R^2$) define a "map of epistasis" in which different mutations may occupy different positions. The edges of this map correspond to limit cases where epistasis is either very strong or very weak, and either entirely global or entirely idiosyncratic (Fig. 1d). In our system, and when there is no drug in the environment, we observed that the strength of epistasis is similar for all mutations ($\text{var}\,\Delta f_i/\text{var}\,f(B)\sim 1$ for all four), but the degree to which this epistasis is global exhibits substantial variation ($R^2\sim 0.7$ for mutation C59R to $R^2\sim 0.2$ for mutation S108N).

### Drug dose modulates the strength and shape of global epistasis

To analyze how the environment may modulate epistasis in our system, we considered every concentration of pyrimethamine ranging from $10^{-2}\,\mu$M to $10^3\,\mu$M, as well as the no drug control, in the dataset described above (Fig. 2a). We found that the strength and shape of global epistasis varied substantially for all mutations as the concentration of pyrimethamine increased. As an illustration, in Fig. 2b we show that mutation C59R goes from exhibiting a pattern of diminishing returns (smaller fitness effects in higher-fitness backgrounds) at low drug doses to exhibiting increasing returns (larger positive fitness

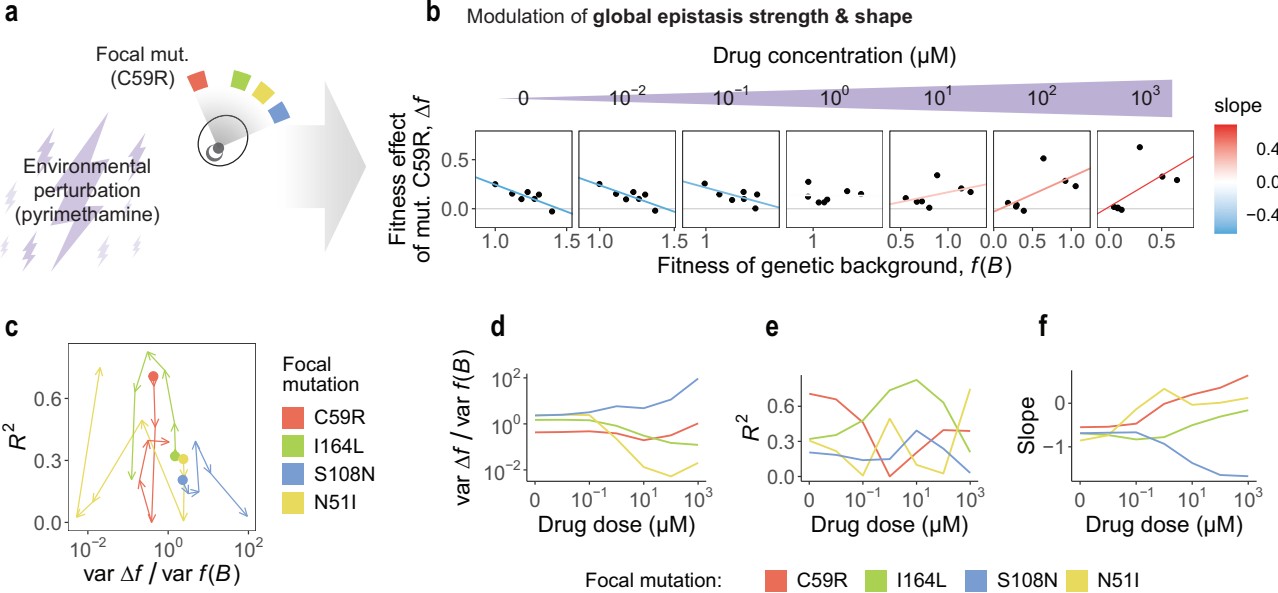

**Fig. 2 | The environment modulates the strength and shape of global epistasis. a** We analyzed how an environmental perturbation (in the form of a pyrimethamine concentration gradient) affects the global epistasis patterns observed in our four-mutation landscape. **b** As an example, we show that increasing drug dose alters the global epistasis pattern for mutation C59R, going from diminishing returns to increasing returns. **c** Drug concentration affects epistasis differently for each

mutation, which can be thought of as mutations following different paths in our "epistasis map". Arrows follow the direction of change as drug dose increases from 0 to $10^3\,\mu$M. **d–f** The strength of epistasis (variance ratio $\text{var}\,\Delta f/\text{var}\,f(B)$), the degree to which epistasis is global ($R^2$) and the shape of global epistasis (slope of the regression between $\Delta f$ and $f(B)$) are modulated differently for each mutation.

effects in higher-fitness backgrounds) at high doses. For other mutations, epistasis is modulated differently (Supp. Figure 2).

Mutations can be seen as describing a "path" through the map of epistasis we defined above (Fig. 2c) as the environment changes, i.e., as drug concentration increases. For instance, mutation S108N becomes largely idiosyncratic, moving to the bottom-right corner of the map. Epistasis for mutation N51I becomes weaker as drug dose increases, i.e., this mutation describes a path towards the left side of the map. For mutations C59R and I164L, the strength of epistasis remains roughly constant across drug concentrations (variance ratio var $\Delta f_i$/var $f(B) \sim 1$), but epistasis becomes more global (higher $R^2$) for I164L and more idiosyncratic (lower $R^2$) for C59R at intermediate drug doses. Beyond modulating the variance ratio (Fig. 2d) and the $R^2$ (Fig. 2e), drug dose also changes the shape of global epistasis. Interestingly, while the slopes of the linear regressions between $f(B)$ and $\Delta f$ are similar for all mutations at low doses, they become substantially different at higher concentrations (Fig. 2f). Note that these three quantities (variance

## BOX 1

# Effective genetic interactions

Recent work has demonstrated that global epistasis patterns can emerge as a result of just a few[14], or multiple widespread[48], gene-by-gene interactions. In the latter case, a recent study has found an explicit quantitative relationship connecting the shape of global epistasis to pairwise and higher-order interactions between mutations[48]. Theory has suggested that this relationship may hold even when such interactions are sparse as opposed to widespread[49]. In particular, these works show that the global epistasis slope of a focal mutation $i$ (denoted $b_i$) can be approximated by a sum of contributions from every other mutation $j \neq i$ as[48,49]:

$$b_i \approx \sum_{j \neq i} \omega_{ij} \beta_{ij} \tag{1}$$

where $\omega_{ij}$ and $\beta_{ij}$ are defined as

$$\beta_{ij} \equiv \frac{\langle \epsilon_{ij} \rangle}{\langle \Delta f_j \rangle} \tag{2}$$

$$\omega_{ij} \equiv \frac{\langle \Delta f_j \rangle^2}{\sum_{k \neq i} \langle \Delta f_k \rangle^2} \tag{3}$$

The term $\langle \Delta f_j \rangle$ represents the average fitness effect of mutation $j$ across all backgrounds that do not carry mutations $i$ nor $j$. The term $\langle \epsilon_{ij} \rangle$ represents the average pairwise epistasis between mutations $i$ and $j$, defined as the deviation between the fitness of the double mutant with respect to the expectation that mutations $i$ and $j$ do not interact.

As an illustration, the figure in this Box shows the case where the genetic background corresponds to the genotype carrying only mutation N51I. In the illustration, mutation $i$ is I164L and mutation $j$ is C59R, which has a positive fitness effect ($\Delta f_j > 0$) in this background. When both mutations are introduced, the resulting fitness is lower than the addition of the two separate fitness effects, indicating negative epistasis ($\epsilon_{ij} < 0$). The terms $\langle \Delta f_j \rangle$ and $\langle \epsilon_{ij} \rangle$ can be computed by quantifying $\Delta f_j$ and $\epsilon_{ij}$ in every possible genetic background which does not carry any of the two mutations, and averaging them.

The expressions above result from writing the variances of $\Delta f$ and $f(B)$, as well as the covariance between the two magnitudes, in terms of the presence or absence of each mutation[48]; and then truncating those expressions at the lowest-order terms (see refs. 48 & 49 and Supplementary Notes for a more detailed derivation). This is approximation partially neglects the effect of higher-order interactions (HOIs): while, in principle, HOIs can make the magnitude of pairwise epistasis ($\epsilon_{ij}$) vary depending on the presence/absence of additional mutations, this variation is averaged across all backgrounds (as $\langle \epsilon_{ij} \rangle$) in equations 2 and 3. If we did not explicitly neglect higher-order terms, equations 1–5 would contain additional terms depending on $\langle \epsilon_{ijk} \rangle$ (defined as the average deviation in fitness of the triple mutant with respect to the additive expectation from each of the three single mutants), $\langle \epsilon_{ijkl} \rangle$, etc.

(Supplementary Notes). The larger the magnitude of such terms, the more substantial the deviation we will observe between the empirically measured variance ratio, $R^2$, and slope and the values estimated by equations 1-5.

We propose that the term $\beta_{ij}$ may be interpreted as the quantification of an "effective interaction" between mutations $i$ and $j$. In turn, $\omega_{ij}$ can be seen as a "weight" which captures the fraction of fitness variance due to the presence/absence of mutation $j$ across the genotypes that do not carry mutation $i$[49]. Thus, equation 1 indicates that the global epistasis slope for mutation $i$ ($b_i$) can be estimated as the weighted first moment (i.e., the weighted mean) of the distribution of effective interactions for that mutation: $b_i \approx \sum_{j \neq i} \omega_{ij} \beta_{ij} = \langle \beta \rangle_\omega$.

Extending this theoretical framework (Supplementary Notes), one can readily derive that the variance ratio (var $\Delta f$/var $f(B)$) and the $R^2$ for a focal mutation $i$ can be approximated respectively as

$$\frac{\text{var } \Delta f_i}{\text{var } f(B)} \approx \sum_{j \neq i} \omega_{ij} \beta_{ij}^2 = \underbrace{\langle \beta^2 \rangle_\omega}_{\substack{\text{Weighted 2nd moment} \\ \text{of } \beta}} \tag{4}$$

$$R_i^2 \approx \frac{\left( \sum_{j \neq i} \omega_{ij} \beta_{ij} \right)^2}{\sum_{j \neq i} \omega_{ij} \beta_{ij}^2} = \frac{1}{1 + \text{CV}_\omega^2(\beta)} \tag{5}$$

where $\text{CV}_\omega$ is the weighted coefficient of variation of the distribution of effective interactions, i.e., its weighted standard deviation relative to its weighted mean.

These effective interactions result from the composition of two contributions to the pattern of global epistasis for a given mutation. This pattern emerges in the form of a relationship between the fitness effect $\Delta f$ of the mutation (represented on the y-axis) and the background fitness $f(B)$ (represented on the x-axis). Intuitively, the terms $\langle \epsilon_{ij} \rangle$ generate "vertical" variation, making the fitness effect of mutation $i$ vary depending on the presence or absence of mutation $j$. The terms $\Delta f_j$ generate "horizontal" variation in fitness across backgrounds that carry or do not carry mutation $j$. What we here call an "effective interaction" $\beta$ (equation 2) and a "weight" $\omega$ (equation 3) are just convenient rearrangements of these terms that allow us to approximate the variance ratio, $R^2$, and slope in terms of the weighted moments of a common distribution.

From equation 3, one might expect that considering additional mutations beyond the four we analyze here will tend to decrease the magnitude of most individual weights (as it will add more terms to the denominator). This, however, does not mean that the overall strength of global epistasis will necessarily decrease. Whether this is the case will ultimately depend on how the focal mutation interacts with the additional loci. Considering additional mutations increases the number of possible genetic backgrounds over which the terms $\langle \Delta f_j \rangle$ are

averaged, and it also increases the number of terms in the sums in equations 1, 4, and 5. Thus, mutations may exhibit strong global epistasis even in very large combinatorial fitness landscapes[14].

ratio, $R^2$, and slope) are not independent, as they satisfy $b^2 = R^2 \operatorname{var}\Delta f/\operatorname{var}f(B)$ (where we have denoted the slope as $b$).

Using cycloguanil instead of pyrimethamine as the environmental perturbation modifies the paths that mutations follow in the epistasis map, but some qualitative behaviors are similar for both drugs: for instance, epistasis for mutation S108N also becomes largely idiosyncratic at high cycloguanil doses, while epistasis for mutation I164L becomes strongly global at intermediate doses (Supp. Figures 3 and 4). Together, these observations highlight the role of the environment in dictating the strength and shape of global epistasis in this particular system.

**Environmental modulation of global epistasis is explained by changes in background fitness effects and interactions**

Can we attribute these environmental effects on global epistasis to specific interactions between the four mutations in this landscape? Recent theory has shown that the slope of the regression between $\Delta f$ and $f(B)$ for a given mutation can be quantitatively explained in terms of fine-grained genetic interactions between that mutation and every other loci in its genetic background[48,49] (equations 1 to 3 in Box 1). Here we show that the strength of epistasis (variance ratio $\operatorname{var}\Delta f/\operatorname{var}f(B)$) and the degree to which epistasis is global ($R^2$ of the $\Delta f$-vs-$f(B)$ regression) can also be quantitatively linked to specific gene-by-gene interactions (Box 1 and Supplementary Notes).

For that, we define the "effective interaction" between mutations $i$ and $j$ (denoted $\beta_{ij}$) as the average magnitude of epistasis for those mutations (i.e., the average deviation between the fitness of a genotype carrying both mutations with respect to the additive expectation that they do not interact, see Box 1) with respect to the average fitness effect of mutation $j$ across all genetic backgrounds of $i$ (see equation 2 in Box 1). Expanding on previous theoretical results, in Box 1 we show that the variance ratio, $R^2$, and global epistasis slope for a focal mutation $i$ can all be quantitatively linked to the mean and standard deviation of the distribution of that mutation's effective interactions (see also Supplementary Notes for a more detailed derivation of the expressions in Box 1).

In light of equations 1 to 5 (Box 1), we may expect environmental factors to modulate the strength and shape of global epistasis for a focal mutation through two different mechanisms: either by modifying the fitness effects of the background mutations, or by modulating the interactions between them and the focal (Fig. 3a, b). To test these two potential mechanisms, we quantified the terms $\Delta f_j$ and $\epsilon_{ij}$ for every pair of mutations $i$ and $j$ in the dataset, in every possible genetic background and for every drug concentration between 0 and $10^3$ μM. We started by analyzing the case with C59R as the focal mutation $i$,

considering the other three mutations as non-focal background mutations ($j$).

We quantified the fitness effects $\Delta f_j$ of each non-focal mutation in every genetic background not carrying C59R (see Box 1). As shown in Fig. 3a, drug concentration modulates these fitness effects differently for each background mutation. For instance, mutation S108N has a very small average fitness effect ($\langle\Delta f_{S108N}\rangle \sim 0$) at low doses, which becomes strongly positive ($\langle\Delta f_{S108N}\rangle > 0$) as the concentration of pyrimethamine increases in the environment (Fig. 3a, blue). On the other hand, mutations I164L and N51I have small and negative average fitness effects ($\langle\Delta f_{I164L}\rangle$, $\langle\Delta f_{N51I}\rangle < 0$) at low doses, but the fitness effect of N51I becomes slightly positive ($\langle\Delta f_{N51I}\rangle > 0$) at high doses (Fig. 3a, yellow).

We then analyzed the average magnitude of epistasis ($\langle\epsilon_{ij}\rangle$, quantified as explained in Box 1) between the focal mutation C59R and each of the three background mutations. Again, we found that the environment modulates these epistatic effects differently for each background mutation (Fig. 3b). For instance, mutation S108N goes from exhibiting low positive epistasis with C59R ($\langle\epsilon_{C59R,S108N}\rangle > 0$) at low doses to exhibiting very large positive epistasis ($\langle\epsilon_{C59R,S108N}\rangle >> 0$) with C59R at higher concentrations (Fig. 3b, blue). In turn, mutation N51I exhibits, on average, moderate positive epistasis with the focal mutation C59R ($\langle\epsilon_{C59R,N51I}\rangle > 0$) at low drug doses, and moderate negative epistasis ($\langle\epsilon_{C59R,N51I}\rangle < 0$) at high doses (Fig. 3b, yellow).

Using equations 2 and 3 in Box 1, we then quantified the magnitude of the effective interaction ($\beta$) between C59R and each of the three non-focal mutations for every drug dose. We also quantified the weight ($\omega$) corresponding to each background mutation. In Fig. 3c we show how the distribution of effective interactions of C59R changes as pyrimethamine concentration increases. Most notably, at low doses the effective interactions of C59R with I164L (green dots in Fig. 3c) and N51I (yellow dots in Fig. 3c) have the largest weights, and are negative as a result of $\langle\epsilon_{ij}\rangle$ and $\langle\Delta f_j\rangle$ having opposite signs for these two mutations. As drug concentration increases, however, the weight of the positive effective interaction between C59R and S108N (blue dots in Fig. 3c) becomes larger, since the average fitness effect of S108N increases substantially.

As explained in Box 1, the (weighted) first and second moments of the distribution of a mutation's effective interactions modulate the strength and shape of global epistasis for that mutation. In particular, the weighted mean estimates the global epistasis slope (equation 1), the weighted second moment estimates the variance ratio $\operatorname{var}\Delta f/\operatorname{var}f(B)$ (equation 4), and the coefficient of variation estimates the $R^2$ (equation 5). In Fig. 3d–e we show that for mutation C59R, equations 1 to 5 do a good job at capturing how the slope and variance ratio change across drug concentrations. For the $R^2$ (Fig. 3f) the quantitative agreement is weaker, but the qualitative behavior is well captured –

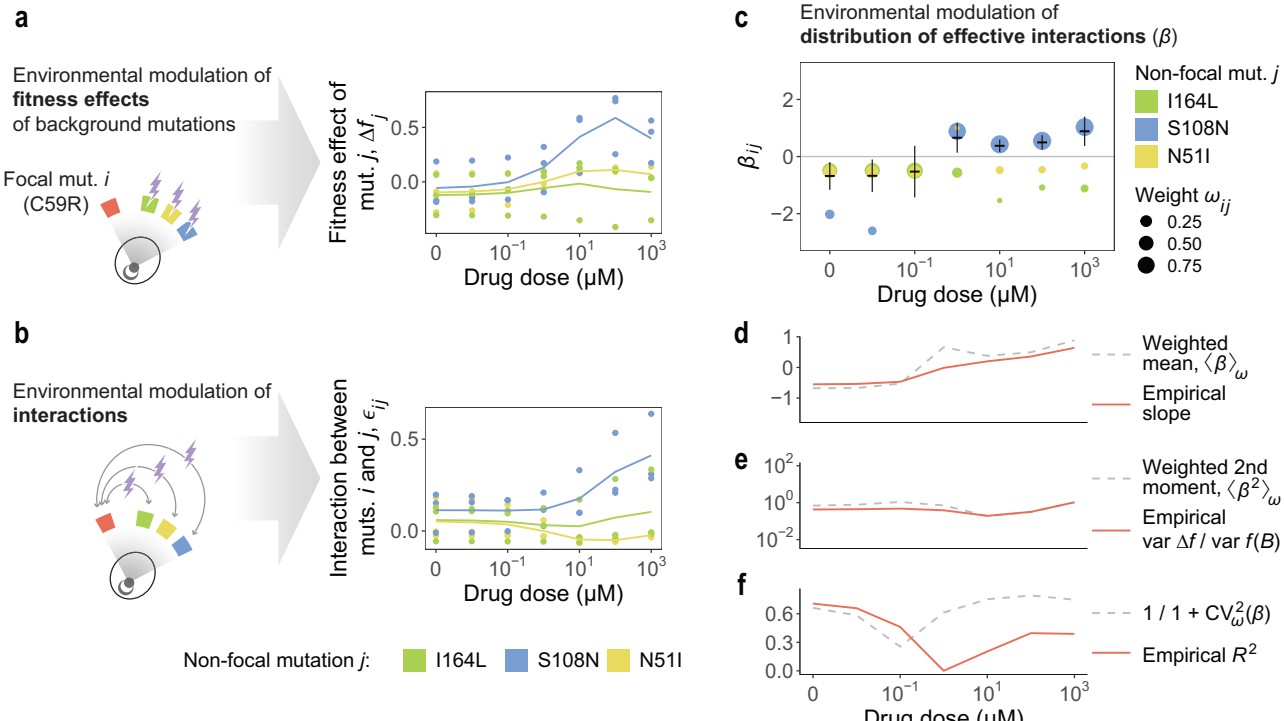

**Fig. 3 | Effective interactions explain the effect of the environment on global epistasis for mutation C59R. a** We quantified the fitness effect of mutations I164L (green), N51I (yellow) and S108N (blue) on each background not carrying the focal mutation (here C59R) and for every pyrimethamine dose. Each dot corresponds to a different background. Lines represent means across all backgrounds. **b** In every background not carrying mutation C59R, we also quantified the magnitude of epistasis between C59R and each of the three background mutations, i.e., the deviation between the fitness of the double mutant with respect to the additive expectation (see Box 1). **c** Distributions of effective interactions (as defined in Box 1) for mutation C59R at different drug concentrations. Each color corresponds to the effective interaction of C59R with each of the other three mutations. Dot size is proportional to the weight of that effective interaction (as defined in equation 3). Black lines and error bars represent weighted means and weighted standard deviations across the $n = 3$ background mutations. **d**–**f** As drug dose increases, the moments of the distribution change (dashed lines), as do the strength of epistasis (variance ratio var $\Delta f$/var $f(B)$), the degree to which epistasis is global ($R^2$), and global epistasis slope for the focal mutation C59R (solid red lines).

namely, the $R^2$ being high at low and high doses, and dropping at intermediate concentrations.

This analysis allows us to identify the mechanisms underlying the environmental modulation of global epistasis for this mutation. As we had shown in Fig. 2b, C59R exhibits a pattern of diminishing returns at low drug doses and a pattern of increasing returns at high doses. This can be explained by the negative effective interactions of C59R with I164L and N51I dominating at low pyrimethamine concentrations, and the positive effective interaction between C59R and S108N dominating at high drug concentrations (Fig. 3c).

Considering any of the other three mutations (instead of C59R) as the focal yields a similarly good agreement of the empirically observed variance ratio, slope, and $R^2$ with the values estimated using equations 1 to 5. For every mutation in the dataset, we quantified its distribution of effective interactions with all other mutations at all drug doses. In Fig. 4 we show that the path described by each mutation in the "epistasis map" closely matches the path expected from estimating var $\Delta f$/var $f(B)$ and $R^2$ using the distribution of a mutation's effective interactions.

Equations 1-5 can generally provide accurate estimates for the strength and shape of global epistasis across concentrations of pyrimethamine (Fig. 4b–d) or cycloguanil (Supp. Figure 5). Importantly, however, these estimates sometimes deviate from the empirically observed variance ratio, $R^2$ and slope. This is perhaps most evident in Figs. 3f and 4d (red dots), showing that the $R^2$ for mutation C59R is overestimated by equation 5 at high pyrimethamine doses. As explained in Box 1, equations 1 to 5 result from partially neglecting the effect of higher-order interactions (HOIs) between mutations. Therefore, the deviations observed in this particular case point to mutation

C59R engaging in strong HOIs at high drug doses. Interestingly, these deviations seem to be smaller for the cycloguanil data (Supp. Figure 5), indicating a smaller effect of HOIs in the presence of this drug. Together, these results highlight the utility of defining effective interactions in order to uncover which fine-grained genetic interactions, and through which mechanisms, play a more prominent role in shaping global epistasis across environments.

## Discussion

The observation that epistasis often has a global component has set a stepping stone in our ability to understand and predict adaptation. For instance, diminishing returns epistasis can explain the strikingly conserved patterns of declining adaptability reported in long-term evolution experiments[7–9,50,51]. While there have been increasing efforts to understand how patterns of global epistasis emerge from fine-grained genetic interactions[13,14,48,49,52,53], less attention has been paid to how they might be affected by the environment. This is particularly important in the context of AMR evolution, where the use of antimicrobial drugs can often exert evolutionary pressures that reshape the topography of genotype-phenotype maps and drive the emergence of resistant genotypes.

In this work, we have examined how the concentration of a drug in the environment affects the patterns of global epistasis in a particular fitness landscape defined by four mutations often found in resistant clones of the *P. falciparum* malaria parasite[42–44]. Our results show that the strength of epistasis, the degree to which epistasis is global, and the shape of global epistasis are strongly modulated by drug concentration in this landscape. Furthermore, we have shown that the origins of this modulation can be attributed to

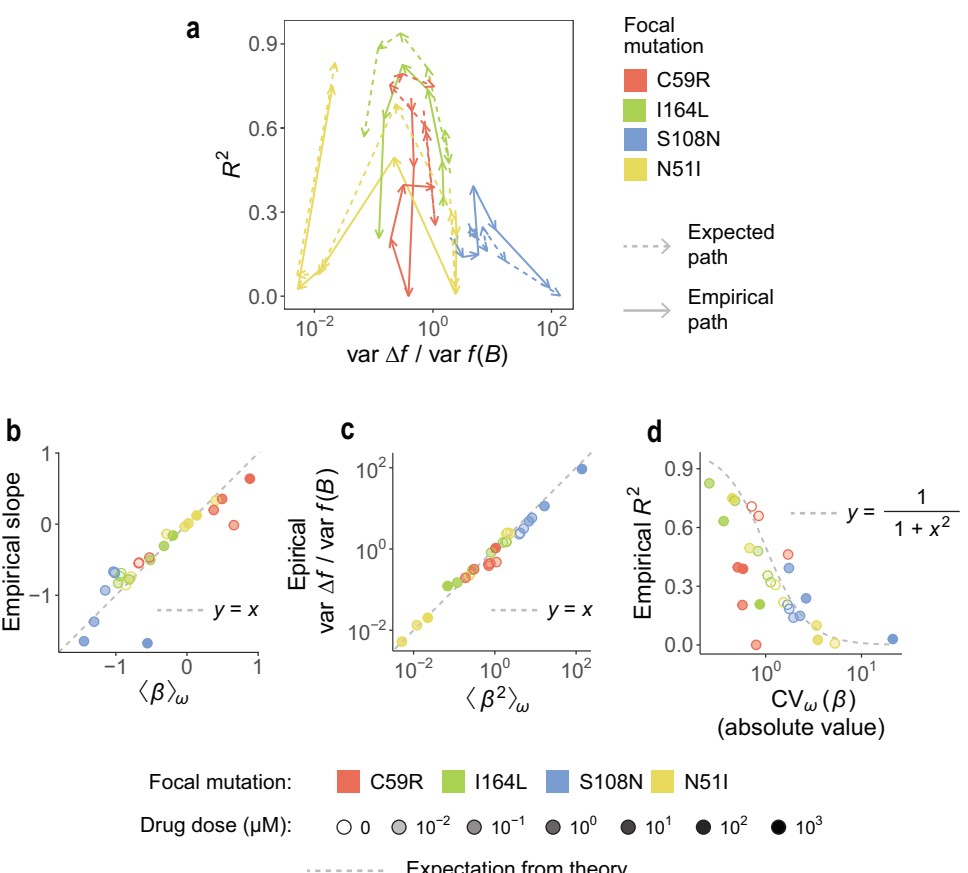

**Fig. 4 | The distribution of effective interactions explains the strength and shape of global epistasis across drug concentrations for all mutations in the system. a** We show the paths that mutations follow in the "map of epistasis" as drug concentration increases. Solid lines: paths obtained from empirically quantifying the variance ratios and $R^2$ (same as in Fig. 2c). Dashed lines: paths obtained by estimating variance ratios and $R^2$ from equations 4 and 5 in Box 1. **b–d** We compare the empirically obtained values for the variance ratio, $R^2$, and global epistasis slope for every mutation in the dataset and at all concentrations of pyrimethamine with the values estimated from each mutation's distribution of effective interactions (equations 1, 4, and 5 in Box 1).

specific gene-by-environment and gene-by-gene-by-environment interactions, expanding on previous theoretical results connecting the emergence of global epistasis to pairwise and higher-order genetic interactions[48,49]. While this connection was originally formulated under the assumption of widespread epistasis[48], theory had suggested that it may hold even when genetic interactions are sparse[49].

Our results provide direct empirical evidence of this theoretical framework successfully explaining global epistasis patterns in a particular drug resistance fitness landscape, even when these are dominated by a single interaction. We have shown that, at high drug doses, the strength and shape of global epistasis for mutation C59R is dictated by its interaction with a single background mutation (S108N). This is consistent with recent work demonstrating that global epistasis can emerge as a result of sparse genetic interactions[14]. More generally, we have shown that the collective action of all mutations' epistatic and fitness effects determines the shape of a distribution of "effective interactions," which we have defined mathematically building on recent theory from quantitative genetics. We have demonstrated that the moments of this distribution determine the properties of global epistasis for a given mutation − namely, its strength and the shape of the relationship between the background fitness and the mutation's fitness effect.

Here we analyzed a low-dimensional landscape, which allowed us to directly test the fitness effect of every mutation, as well as every possible epistatic coefficient between all pairs of mutations, in almost every genetic background. All mutations we considered correspond to a same gene (encoding the DHFR enzyme of *P. falciparum*), and thus we might expect interactions between them to be stronger than with other potential mutations in more distal sites of the genome. Further work will be required to determine whether effective interactions can be reliably defined and quantified in higher-dimensional landscapes, including mutations in different genes or pathways, where empirically measuring the fitness of every genotype across environments might be more challenging. For instance, the contribution of each background locus to the global epistasis pattern of a focal mutation can be expected to become smaller (lower weight $\omega$ as defined in equation 3) as the size of the landscape increases. Whether this results in stronger or weaker global epistasis patterns will generally depend on the structure of epistasis among all mutations considered, which can be expected to vary across landscapes of different sizes. In addition, here we have focused on two specific drugs as our environmental variables. It will be important to test whether different forms of environmental perturbations (e.g., temperature or pH variations), and within what ranges, have the potential to modulate global epistasis in this landscape and others. Most importantly, our work suggests that the global epistasis framework can be readily extended to account for environmental variables, which might be critical to predict adaptive trajectories under changing environments in the context of AMR evolution.

## Methods
### Data analysis
Data was obtained from the original publications[31,34,41]. The genotype carrying mutations I164L and S108N was removed from the data due to having inconsistent fitness values across two no-drug controls. All analyses were carried out using R version 4.3.1.

## Quantification of effective interactions

For a given pair of mutations $i$ and $j$, we denote $B$ a genetic background not carrying any of the two. We denote $B+i$ and $B+j$ the genotypes resulting from mutating $i$ or $j$, respectively, in the background $B$. The double mutant is denoted $B+i+j$. To compute the effective interaction $\beta_{ij}$ (see equation 1 in Box 1), we consider every possible background $B$ and quantify

$$\Delta f_j(B) = f(B+j) - f(B) \tag{6}$$

$$\epsilon_{ij}(B) = f(B+i+j) - f(B+i) - f(B+j) + f(B) \tag{7}$$

We then compute $\langle \Delta f_j \rangle$ and $\langle \epsilon_{ij} \rangle$ by averaging the expressions above across every possible genetic background $B$. If any of the terms on the right-hand side of the equations above were missing (due to incomplete data), we computed the averages across the known terms only.

## Reporting summary

Further information on research design is available in the Nature Portfolio Reporting Summary linked to this article.

## Data availability

The data analyzed in this study were obtained from the original publications[31, 34,41] and is also available at https://github.com/jdiazc9/env_global_epist[54]. Code used to generate all figures can be found in the same GitHub repository.

## Code availability

All code is available at https://github.com/jdiazc9/env_global_epist[54].

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

## Acknowledgements

We thank members of the Sanchez and Ogbunu labs for helpful discussions. C.B.O. acknowledges support from National Science Foundation's Division of Environmental Biology Award Number 2142720 and from the MLK Visiting Scholars and Professors Program at the Massachusetts Institute of Technology. J.D.-C. & A.S. were partially supported by grant PID2021-125478NA-100 funded by MCIN/AEI/10.13039/501100011033 and by "ERDF: A way of making Europe."

## Author contributions

J.D.-C., A.S. and C.B.O. conceived the study. J.D.-C. analyzed and interpreted data. J.D.-C. wrote the paper, with input from A.S. and C.B.O.

## Competing interests

The authors declare no competing interests.
