## [Peer Review File · Nature Communications]

Environmental modulation of global epistasis in a drug resistance fitness landscapeReviewers' Comments:

Reviewer #1:

Remarks to the Author:

In 'Environmental modulation of global epistasis is governed by effective genetic interactions', Diaz-Colunga and colleagues introduce us to a new way to quantify (and conceptualize/ascribe causality) to genotype-phenotype maps that change in the face of a continuously varying environmental background. Taking the fitness landscape $G \times E$ concept one step further into the world of 'seascapes'.

Overall, I find this to be a clearly written, well explained, thoughtful and rigorous analysis of a new phenomenon. I think that it will help move the field forward in a meaningful way, but felt somewhat unsatisfied by some of the parsimony of the analysis -- in particular, while I understand that for clarity, focusing the results and observations on one 'focal' mutation is simplifying, i think it misses an opportunity to highlight more holistic, global patterns and permit broader characterization and comparison, and possibly to extend our understanding.

Broadly, I think that there could be an attempt to condense the analysis so that we can see 'all the data' in one fell swoop. For example, could figure 1D be condensed from a series of dot plots with changing dose, to a single dot plot with dose on the x-axis and the derivative of the fitness effect on the y-axis? This would give us a larger view of the effect of the changing environment for this focal mutation, and would then admit the chance to plot all of the other vantage points (focal mutations) in the same plot, giving us a chance to see the global patterns (as advertised). I would be excited to see how this would elucidate the changes more broadly, and to see what further interesting questions this might raise.

I will hold off on providing any other specific comments on the results or discussion section because if the authors choose to attempt the analysis suggested above, i think those sections will naturally be somewhat rewritten/strengthened.

I will, however give some specific feedback on the figures and some of the mathematical exposition. Some of these comments are stylistic, though I think could aid in clarity.

Figure 2 - general comment: Might this figure be broken into two figures? the bottom panel is really just a cartoon to help us understand the results we see, while the top is some data -- exemplar data i understand -- but data nonetheless. should this be in the discussion (or intro)? as a stand alone figure, the bottom panel could be decorated with some mathematical exposition to help us understand how the model is constructed. some journals have 'box' formatting tools to help us understand model construction, this might be a nice way to separate out this subfigure and the mathematical exposition all in one place.

A: somewhat hard to interpret as drawn in my opinion. could the authors represent this as a hypercube, and color the nodes by fitness and use directed edges? I think this would be easier to parse. the caption for panel A doesn't explain the 'special' 2-cube in the data (panel b's caption does...)

equations -- as above, i think putting all the maths and the cartoon into a box would be useful. further, i think equation 1 is redundant, and it would be clearer to just show 2 and 3. nitpicking here, but in equation 1 the use of the 'x' symbol -- this is just meant to be multiplication, yes? maybe simpler to leave the symbol out so as not to confuse for vector or cross product.

figure 3 is nice, but again, i'd really like to see 'global' patterns as advertised in the title, rather than the arbitrary observation of a single focal mutation. i'm not 100% sure how to represent these data for all...

As to the minimal model -- some of the results seem to follow by construction, and again, i think we are missing an opportunity to learn something about this exciting phenomenon. A couple ideas that wouldn't require that much extra work, but could significantly improve our understanding: there are now several other groups (including ours) that have reported empirical seascapes in bacterial species under antibiotic stress (Mira et al in MBE and Das et al in eLife in addition to our bioRxiv preprint). I would love to see where, in the space of the parameters of the minimal model, these seascapes lie. Then, a cool opportunity exists to bootstrap some random seascapes and see where in that space those lie. It could be that empirical seascapes can only exist in certain part of this parameter space. Depending on the how one bootstrapped these random seascapes, you could also study how evolutionary (dose dependent) fitness tradeoffs effect this -- with no tradeoff (0 slope in something like your figure 1A) being being a null model, a negative slope being the most physical, and possibly a positive slope being non-physical/unobserved. This latter idea could be outside the scope of the current work, but is an interesting avenue of thought I'd be curious to hear the authors opinions on.

A few final thoughts: i think a more thorough explanation of the minimal model is important for readers to better understand what is going on. The reader is currently referred to previous work from the authors (see ref 44 we are told). While this is fair, I think at least a bit more explanation would be helpful.

In the supplements, some of the linear fits are pretty awful... maybe reporting the correlations could elucidate something important -- it might be that where the linear model breaks down something else is going on. I'm not 100% sure how to best represent this, but I think it deserves some thought or at a minimum, reporting.

Overall, I am extremely excited but this work, and am eager to see it in print. I do think that finding a way to represent all the data at once would be helpful, and that using this analytical framework on other available seascapes could be enlightening. Kudos to the authors for this creative and enlightening work, and thank you for the opportunity to review it and offer opportunities to improve its impact.

best,

jake scott

Reviewer #2:

Remarks to the Author:

Global epistasis is a phenomenon where the fitness effects of mutations across genetic backgrounds can be described with few parameters, for example, a negative linear dependence on background fitness. This phenomenon has been observed in numerous microbial systems and recent theoretical work has proposed various hypotheses to explain its origin. In this paper, the authors examine the variation in global epistasis patterns using a previously published dataset of a 4-site fitness landscape measured across drug concentrations in the *P. falciparum* parasite.

While examining the variation of global epistasis patterns across environments is an important goal and the analysis itself is sound, I find the new insights brought by this work to be incremental w.r.t past work. Many of the points made here have been provided in other theoretical works, including the authors' own work (refs 43,44,47). A much larger dataset with a 10-site fitness landscape across 6 environments is available (ref 14) and has been used to explain the origin of these patterns. In my opinion, the community will indeed benefit from extensively re-examining published landscapes in the light of new theory. However, the scope of the analysis presented in this work is limited.

I expand on other concerns below:

1. Global epistasis refers to cases where the fitness effect has a simple linear dependence on background fitness. This is perhaps true for the first four concentrations (panels) in Figure 1D. This dependence certainly does not look linear for the latter three concentrations. The lack of linearity is also apparent for the other focal mutations (Figure S2). Given this, I am uncertain how the mathematical analysis of the change in the slope across environmental conditions should be interpreted. Of course, equation 1 gives a general mathematical expression for the best fit slope when the fitness effect of a mutation is plotted against background fitness. This expression will hold true regardless of whether the data looks linear or not, and the fact that it can explain the best fit “slope” is not surprising. Importantly, this analysis does not explain why linear global epistasis patterns appear or do not appear in the first place.

2. Error in fitness measurements can create spurious negative trends because the error in the estimate of the fitness effect will be anti-correlated with the error in background fitness. To account for this, the authors should include error bars for the fitness estimates/calculate the contribution to the slope made by this anti-correlation.

3. I am puzzled by the value provided by the minimal model surrounding Equation 4 and the paragraphs after line 157. Since the fitness measurements of all 2^4 mutations are available, I presume one can simply fit all the fitness landscape parameters (and perhaps apply a threshold to remove the near-zero terms and obtain the simpler model). However, since these parameters are obtained by fitting to the data, they will of course reproduce the empirical patterns. And since Equation 1 is a general mathematical expression for the slope, changing the parameters of the fitness landscape should lead to slopes of different signs. As stated in point 1 above, the change in slopes does not imply that this model leads to linear global epistasis patterns.

Reviewer #3:

Remarks to the Author:

This manuscript takes advantage of previously published data and theory to present an analysis of how global epistasis among four mutations in *P. falciparum* genes is modulated by the environment (antimalarial drugs). The key result is a switch in the direction of global epistasis – from a negative slope to a positive slope – as the drug concentration increases (Fig 1D). The authors employ an existing model to partition these effects into two parts (background fitness and genetic interactions), which when summed together, approximate the real slope data (Fig 3). They then introduce a new, simple model (Equation 4) parametrized with real data, which is claimed to also reproduce the empirical results.

This manuscript is very well-written. I am moderately familiar with the epistasis literature, but it is not my primary area of expertise. I nonetheless found the mathematical explanations clear and easy to follow. I also appreciated the diagrams in Fig 2.

The scientific context and citations provided in the introduction and discussion suggest that the general problem -- modulation of global epistasis by environmental variables – is important to the larger goal of predicting adaptation. This paper contributes to understanding this problem by presenting a fully-analyzed, concrete example of the modulation of global epistasis by environmental variables. More specifically, this paper shows that existing epistasis theory holds true in the context of sparse interactions. In this way it appears experimentally similar to Hall et al., 2019 (Evolution), but the mathematical treatment here is more thorough. In general, I think this work makes a notable contribution to the stated problem through its comprehensive treatment of the fitness effects (and interactions) of four mutations in multiple drug environments.

Most of the claims in the paper are convincingly supported by the data and models (e.g. Fig 3C and

Fig 1D). The main exception, to my eye, was the claim about the accuracy of the simple minimal model (Fig 4). Although it does reproduce the expected slopes, the positions of the points themselves differ substantially between Fig 4E and Fig 4F. Specifically, the example presented in the text (lines 173-179) notes that the filled squares move to the right on the x-axis with increasing drug concentration (Fig 4E). However, in Fig 4F, the filled squares actually move left on the x-axis. Additionally, the points representing 'both S and N' are consistently far apart in F despite being nearby in E. Does this mean that mutation I does have a substantial effect that is glossed over in the simplified model? More broadly, what is the benefit of using this simple (but seemingly less accurate) model instead of the more accurate model presented in Fig 3? I find the apparent inaccuracy surprising, since the simple model was parameterized with real data. Perhaps it indicates that this simple model is too simple?

Besides additional justification for (or improvements on) this simple model, I have no suggestions for additional experiments or analysis. I find this work to be a thorough and useful analysis of the stated general problem.

Minor comments:

- Please name the genes in which the four studied mutations occur.
- Fig 1 and Fig 2: Is this drawing of a bacterium? *P. falciparum* does not have a flagellum nor a single, circular chromosome.
- Fig 1D and elsewhere: How is it determined whether the slopes are significantly non-zero? Can significance be indicated in the figure panels, including in the supplement where a different color scheme is used?
- Fig 3C predicts that the slope of the global epistasis line will be positive at drug concentration 100. In Fig 1D, however, the slope is zero. Is there any explanation for this discrepancy?
- Please define "microscopic genetic interactions." I see that several citations are provided after this term, but I still find it confusing, since "microscopic" has a much more common meaning in biology that is unrelated to epistasis.

Reviewer #1

In 'Environmental modulation of global epistasis is governed by effective genetic interactions', Diaz-Colunga and colleagues introduce us to a new way to quantify (and conceptualize/ascribe causality) to genotype-phenotype maps that change in the face of a continuously varying environmental background. Taking the fitness landscape $G \times E$ concept one step further into the world of 'seascapes'.

Overall, I find this to be a clearly written, well explained, thoughtful and rigorous analysis of a new phenomenon. I think that it will help move the field forward in a meaningful way, but felt somewhat unsatisfied by some of the parsimony of the analysis – in particular, while I understand that for clarity, focusing the results and observations on one 'focal' mutation is simplifying, I think it misses an opportunity to highlight more holistic, global patterns and permit broader characterization and comparison, and possibly to extend our understanding.

We thank the reviewer for his positive characterization of our work. We agree that focusing our analysis on one particular mutation narrowed the scope of our results. In our original manuscript, we thought that illustrating our analysis step-by-step for a particular mutation would make it easier for readers to follow the logic of the paper. But we do see how this might have overly simplified our message.

While we have opted for maintaining the the step-by-step analysis of mutation C59R in our revised manuscript (e.g., the new Fig. 1C or Fig. 2A-B), we now clarify that this serves illustrative purposes only. We now include new figures showing the results of considering each of the four mutations as the "focal" (e.g., Fig. 1D, Fig. 2C-F, or the new Fig. 4). We explicitly demonstrate that our theoretical framework can explain the environmental modulation of global epistasis for all four mutations in the landscape.

Broadly, I think that there could be an attempt to condense the analysis so that we can see 'all the data' in one fell swoop. For example, could figure 1D be condensed from a series of dot plots with changing dose, to a single dot plot with dose on the x-axis and the derivative of the fitness effect on the y-axis? This would give us a larger view of the effect of the changing environment for this focal mutation, and would then admit the chance to plot all of the other vantage points (focal mutations) in the same plot, giving us a chance to see the global patterns (as advertised). I would be excited to see how this would elucidate the changes more broadly, and to see what further interesting questions this might raise.

The plot suggested by the reviewer is now Fig. 2F of the main text. Also note that, following this and other reviewers' suggestions, we have now expanded our original analysis: in the new version of the manuscript, we do not only analyze the shape of global epistasis (i.e., "the derivative of the fitness effect") but also the overall strength of epistasis and the degree to which epistasis is "global." In Fig. 2C-E we show how these different aspects of epistasis are modulated by drug dose for all four mutations.

I will hold off on providing any other specific comments on the results or discussion section because if the authors choose to attempt the analysis suggested above, I think those sections will naturally be somewhat rewritten/strengthened.

Indeed our analysis has now been substantially expanded, and the corresponding figures and results have been modified accordingly. In short, we now study three different aspects of epistasis: (a) how much epistasis there is for a given mutation, that is, how variable the fitness effect of the mutation is depending on the presence/absence of other mutations in its genetic background; (b) to what degree is epistasis "global", i.e., how accurately the fitness effect of a mutation can be predicted from a linear model linking it to the fitness of its genetic background; and (c) what the shape of global epistasis is, i.e., our original analysis on the slope of the linear regression between Δf and $f(B)$.

Originally, we showed that the global epistasis slope could be interpreted in terms of pairwise gene-by-gene "effective interactions." We have now mathematically redefined these effective interactions, and we show that the way in which they are distributed influences the three aspects of epistasis mentioned above. In particular, we show that the mean of the distribution of effective interactions estimates the global epistasis slope (this is equivalent to our main result in the original version of the paper), but also that the second moment of this distribution and its coefficient of variation estimate the strength of epistasis and the degree to which epistasis is "global," respectively.

I will, however give some specific feedback on the figures and some of the mathematical exposition. Some of these comments are stylistic, though I think could aid in clarity.

Figure 2 - general comment: Might this figure be broken into two figures? the bottom panel is really just a cartoon to help us understand the results we see, while the top is some data – exemplar data i understand – but data nonetheless. should this be in the discussion (or intro)? as a stand alone figure, the bottom panel could be decorated with some mathematical exposition to help us understand how the model is constructed. some journals have 'box' formatting tools to help us understand model construction, this might be a nice way to separate out this subfigure and the mathematical exposition all in one place.

As suggested by the reviewer, we have now included Box 1 summarizing the mathematical exposition of our framework. The top panels of our old Fig. 2 are now included in this Box (with the modifications suggested by the reviewer in his next comment). We have also included a Supplementary Text section with a more detailed derivation of the equations in the Box. We appreciate the reviewer's suggestion, as we find that this formatting helps maintain the flow of the main text and more clearly present the theory in the paper.

A: somewhat hard to interpret as drawn in my opinion. could the authors represent this as a hyper-cube, and color the nodes by fitness and use directed edges? I think this would be easier to parse. the caption for panel A doesn't explain the 'special' 2-cube in the data (panel b's caption does...)

We now use the suggested representation in the Box 1 figure. The “special” 2-cube (which is just a particular example of a pair of mutations in a specific genetic background) serves as an illustration of the meaning of $\langle \Delta f_j \rangle$ and $\langle \epsilon_{ij} \rangle$ in equations 1-5 — this is now more clearly explained in Box 1 itself.

equations – as above, i think putting all the maths and the cartoon into a box would be useful. further, i think equation 1 is redundant, and it would be clearer to just show 2 and 3. nitpicking here, but in equation 1 the use of the 'x' symbol – this is just meant to be multiplication, yes? maybe simpler to leave the symbol out so as not to confuse for vector or cross product.

Done — all the math is now presented in Box 1. We have also removed the “x” symbol to avoid confusion.

figure 3 is nice, but again, i'd really like to see 'global' patterns as advertised in the title, rather than the arbitrary observation of a single focal mutation. i'm not 100% sure how to represent these data for all...

In the new version of the manuscript, Fig. 3 provides a step-by-step analysis of mutation C59R, similar to the original figure in the first version of the paper (but expanded to include the new results on the strength of epistasis, etc. as explained above). But we have now added a new Fig. 4, extending the analysis to all four mutations in the landscape. What we show is that our theoretical framework (i.e., equations 1-5 in Box 1) can be used to explain the strength and shape of global epistasis for every mutation in the landscape and at every drug dose.

In summary, we have attempted to address the reviewer's suggestions to report and explain global epistasis patterns for all mutations in our landscape, while maintaining the analysis of a particular “focal” mutation as a way to guide readers through the logic of our arguments.

As to the minimal model – some of the results seem to follow by construction, and again, i think we are missing an opportunity to learn something about this exciting phenomenon. A couple ideas that wouldn't require that much extra work, but could significantly improve our understanding: there are now several other groups (including ours) that have reported empirical seascapes in bacterial species under antibiotic stress (Mira et al in MBE and Das et al in eLife in addition to our bioRxiv preprint). I would love to see where, in the space of the parameters of the minimal model, these seascapes lie. Then, a cool opportunity exists to bootstrap some random seascapes and see where in that space those lie. It could be that empirical seascapes can only exist in certain part of this parameter space. Depending on the how one bootstrapped these random seascapes, you could also study how evolutionary (dose dependent) fitness tradeoffs effect this – with no tradeoff (0 slope in something like your figure 1A) being being a null model, a negative slope being the most physical, and possibly a positive slope being non-physical/unobserved. This latter idea could be outside the scope of the current work, but is an interesting avenue of thought i'd be curious to hear the authors opinions on.

Throughout the manuscript, we argue that global epistasis patterns can be “dominated” by a single interaction between loci — as is the case, for instance, for the interaction between C59R and S108N at high pyrimethamine doses. Our goal with the minimal model was to provide an interpretation for the meaning of this “dominance.” We wanted to show that this “dominance” can be explained in geometric terms: in short, the presence or absence of the second mutation can induce shifts in the background fitness values and the fitness effects of the first mutation, leading to non-zero slopes in the regressions between $f(B)$ and Δf — somewhat similar to the global epistasis patterns reported in Bakerlee et al. (*Science*, 2022), here showing how they may change as the environment is modified in our particular landscape.

Since this goal was not clear for the reviewers, we see that we missed the mark. We have thus opted for removing the minimal model altogether. We think that the new analyses we present in the revised manuscript provide simpler, more interpretable explanations for what the “dominance” of an interaction means. For instance, as we now demonstrate, the weighted average of the distribution of effective interactions for a given mutation estimates the global epistasis slope observed for that mutation. In the case of the interaction between C59R and S108N at high drug doses, we show (in the new Fig. 3C) that this weighted average is largely driven by the effective interaction between those two loci.

As for the analysis of other available seascapes, we entirely agree that it is an exciting direction for future work. As we see it, however, it falls out of the scope of the current paper. Our goal here is to show how global epistasis can be modulated by environmental factors, and how our framework based on the definition of “effective interactions” can help us understand these effects in a particular landscape, analyzed thoroughly and step-by-step. We thus feel that including additional datasets may obscure the goal of the current work.

Regarding the idea of bootstrapping random landscapes, we agree that this type of control can provide valuable information. Following this and reviewer #2’s comments, we have studied the effect of randomizing the pairing between genomes and fitness in our data. For each drug dose, we generated 500 random landscapes by arbitrarily rearranging the mapping between genotypes and measured fitness values. We found that negative correlations between Δf and $f(B)$ were often observed in these random landscapes. This is an unsurprising consequence of regression to the mean: given a bound set of fitness values, if a genetic background corresponds to the fitness maximum, the fitness effect of any mutation will necessarily be negative, whereas if the genetic background is the lowest-fitness allele any mutation will have a positive fitness effect. Importantly, however, we found that the global epistasis patterns observed in our real landscapes are not compatible with the ones seen in these randomization controls (this is now shown in fig. S5 and discussed in a new Supplementary Materials section). This type of randomization control provides a useful baseline for discerning whether the empirically seen patterns of global epistasis capture meaningful information regarding the structure or the fitness landscape, or if instead they emerge as a simple consequence of regression to the mean.

A few final thoughts: i think a more thorough explanation of the minimal model is important for readers to better understand what is going on. The reader is currently referred to previous work from the authors (see ref 44 we are told). While this is fair, I think at least a bit more explanation would be helpful.

This is a fair point, but as mentioned above, since the three reviewers seem to agree that the utility of the minimal model was unclear, we think that removing it is a more reasonable option.

In the supplements, some of the linear fits are pretty awful... maybe reporting the correlations could elucidate something important – it might be that where the linear model breaks down something else is going on. I’m not 100% sure how to best represent this, but I think it deserves some thought or at a minimum, reporting.

We thank the reviewer for this important point, which prompted us to expand our analyses as described above. We now interpret the “goodness of the fit” (quantified as the R^2 of the linear regression) as a metric of the degree to which epistasis is “global.” As the reviewer points out, some mutations seem to exhibit high degrees of epistasis (indicated by the observation that their fitness effects vary substantially across backgrounds), but without displaying a clear correlation between the fitness effect and the background fitness. In these cases, we interpret that epistasis is strong, but largely “non-global” (i.e., “idiosyncratic”). In fact, we observe a variety of behaviors across mutations and drug doses. Some mutations have a very similar fitness effect across all backgrounds (i.e., display low epistasis), some exhibit very large variation in their fitness effects (high epistasis), as well as different degrees of “global” epistasis (different R^2). We now represent this variation in a conceptual

“map of epistasis” (see updated Figs. 1C, 2C, and 4), which allows to readily distinguish across all different behaviors that mutations may exhibit and how they vary across drug concentrations.

Importantly, our new analyses show that the degree to which epistasis is global/non-global can also be explained in terms of our effective interactions. Similar to how the slope can be expressed as the weighted average of a distribution of effective interactions, we show (Box 1 & Supplementary Text) that the R^2 can be related to the weighted coefficient of variation of said distribution (new equation 5).

Overall, I am extremely excited but this work, and am eager to see it in print. I do think that finding a way to represent all the data at once would be helpful, and that using this analytical framework on other available seascapes could be enlightening. Kudos to the authors for this creative and enlightening work, and thank you for the opportunity to review it and offer opportunities to improve its impact.

best,

jake scott

Reviewer #2

Global epistasis is a phenomenon where the fitness effects of mutations across genetic backgrounds can be described with few parameters, for example, a negative linear dependence on background fitness. This phenomenon has been observed in numerous microbial systems and recent theoretical work has proposed various hypotheses to explain its origin. In this paper, the authors examine the variation in global epistasis patterns using a previously published dataset of a 4-site fitness landscape measured across drug concentrations in the *P. falciparum* parasite.

While examining the variation of global epistasis patterns across environments is an important goal and the analysis itself is sound, I find the new insights brought by this work to be incremental w.r.t past work. Many of the points made here have been provided in other theoretical works, including the authors' own work (refs 43,44,47). A much larger dataset with a 10-site fitness landscape across 6 environments is available (ref 14) and has been used to explain the origin of these patterns. In my opinion, the community will indeed benefit from extensively re-examining published landscapes in the light of new theory. However, the scope of the analysis presented in this work is limited.

To better highlight the new insights presented in this work, we have expanded the quantitative framework presented in our manuscript. Previous work (including our own as the reviewer notes) demonstrated that the slope of a linear regression between $f(B)$ and Δf can be linked to pairwise and higher-order interactions between loci, and that a low-order approximation can generally provide a good estimate for this slope. In our revised manuscript, we extend our analysis beyond the slope of the regression: we now show that the overall strength of epistasis for a mutation (quantified as the variance in the mutation's fitness effects across backgrounds) is estimated by the second moment of the distribution of effective interactions of that mutation. We also demonstrate that the degree to which epistasis is "global" (quantified as the R^2 of the linear regression between $f(B)$ and Δf) can be estimated by the coefficient of variation of that distribution. These theoretical results are explicitly presented in the new Box 1 (with a more detailed derivation in a new Supplementary Materials section), and the corresponding empirical tests are presented in the new Fig. 4 of the revised manuscript. We believe these new results help bring to the forefront what we consider to be the main strength of this work: the thorough dissection of how environmental factors shape global epistasis by modulating specific pairwise epistatic interactions in an empirical fitness landscape.

I expand on other concerns below:

1. Global epistasis refers to cases where the fitness effect has a simple linear dependence on background fitness. This is perhaps true for the first four concentrations (panels) in Figure 1D. This dependence certainly does not look linear for the latter three concentrations. The lack of linearity is also apparent for the other focal mutations (Figure S2). Given this, I am uncertain how the mathematical analysis of the change in the slope across environmental conditions should be interpreted. Of course, equation 1 gives a general mathematical expression for the best fit slope when the fitness effect of a mutation is plotted against background fitness. This expression will hold true regardless of whether the data looks linear or not, and the fact that it can explain the best fit "slope" is not surprising. Importantly, this analysis does not explain why linear global epistasis patterns appear or do not appear in the first place.

As the reviewer notes, it is always possible to regress any two variables via a linear model even when their true relationship might be non-linear. However, linearity is not a main point we intended to make in our manuscript. Because we analyze a low-dimensional landscape, our type of data is not the most adequate to identify potential non-linearities in the relationships between $f(B)$ and Δf — note that larger datasets have been used in other works to address this question directly (e.g., Bakerlee et al. 2022).

To address the reviewer's comment, we ran a series of statistical tests of linearity on our data. We tested whether we could reject the hypotheses that (a) the residuals of our linear models were normally distributed (Shapiro-Wilk test for normality); (b) the residuals of the linear models were homogeneously spread across the x-scale (Breusch-Pagan test for heteroscedasticity); and (c) the residuals of the fit were not auto-correlated (ANOVA of linear model fitted to the residual autocorrelation plot). We found that in the large majority of cases, none of the three hypotheses could be rejected ($p > 0.05$ in all three tests): out of our 56 regressions between $f(B)$ and Δf (2 drugs \times 7 doses \times 4 mutations = 56 regressions), in 47 of them ($\sim 84\%$ of the cases) we could not reject the hypotheses that the relationship between the fitness effect and background fitness was linear.

Similarly, we found that a cubic model did not explain significantly more variance than a linear model ($p > 0.05$, ANOVA) in 52 out of the 56 cases ($\sim 93\%$).

The regressions for mutation C59R at high pyrimethamine doses correspond to cases that do not pass the linearity tests, as indicated by the reviewer. This is in fact an interesting situation. The non-linearity arises because points cluster into two different groups, which correspond to genetic backgrounds that carry or do not carry mutation S108N, respectively (see figure below, left panel). Each group thus defines a different “region” of the fitness landscape. Within each region, linear regressions seem to work well (though, of course, the quality of the fits cannot be reliably quantified based on just 3-4 observations). When we try to fit all data using a same model for both regions, we find that a non-linear model is more appropriate than a linear model in this particular case — see the right panel below, where the dashed red line corresponds to a hypothetical non-linear regression between $f(B)$ and Δf which emerges when considering both regions (but note that this is just an illustration and is not intended to convey that the red line reflects the true functional form of the relationship between $f(B)$ and Δf).

This observation opens up many interesting questions: when can we expect to observe this “regionality”? Are regressions truly linear within each region? What determines whether different regions will aggregate into a single “global” trend, and when will this trend be truly linear? While these are all important points, they fall out of the scope of our current manuscript, and again the low dimensionality of our dataset make it inadequate to study them. In Bakerlee et al. these questions are more directly addressed — notably, that study reports similar “regional” patterns for many mutations within a much larger combinatorial space than the one in our paper. Yet, the low dimensionality of our dataset, together with the observation that the hypothesis of linearity most often cannot be rejected, make it difficult to justify systematically imposing more complex, non-linear models on our data. We think that linear regressions are the more parsimonious approach in this case.

There is, however, the question of *how well* a linear model may explain the relationship between $f(B)$ and Δf — specifically, how well a linear model predicts the fitness effect of a mutation with respect to the “naive” prediction provided by the average fitness effect of that mutation, $\langle \Delta f \rangle$. We agree with the reviewer that this is an important point which was not considered in our original manuscript. We have now directly addressed this question in our revised manuscript. We now quantify the R^2 of the linear regressions between $f(B)$ and Δf , and study how this R^2 changes as drug dose increases. We interpret a high R^2 as an indication that epistasis is largely “global.” On the other hand, a low R^2 is interpreted as epistasis being largely “non-global” — for instance, in those cases where there might be no relationship between $f(B)$ and Δf , or in situations where this relationship might be globally non-linear even if it might be locally linear, such as the one shown in the figure above.

Importantly, in our revised manuscript we show that the R^2 can be written in terms of the coefficient of variation of the distribution of effective interactions we defined mathematically. Equation 5 in the new Box 1 shows that the R^2 can be expected to be large when the distribution of effective interactions for a mutation has a low coefficient of variation and vice-versa. This is empirically tested in the new Fig. 4D. We believe this new results further emphasize the utility of the quantitative framework in our paper. We have also substantially revised the main text, attempting to clarify that our analysis is aimed to understand how well a linear model may predict the fitness effect of a mutation, but avoiding any claim that the relationship between $f(B)$ and Δf need always be truly linear.

2. Error in fitness measurements can create spurious negative trends because the error in the estimate of the fitness effect will be anti-correlated with the error in background fitness. To account for this,

the authors should include error bars for the fitness estimates/calculate the contribution to the slope made by this anti-correlation.

As the reviewer points out, spurious anti-correlations between $f(B)$ and Δf can be observed as a simple consequence of regression to the mean. This is observed, for instance, when the mapping between genotypes and fitness is random (i.e., in a maximally “unstructured” landscape). Biases in measurement error can also produce spurious anti-correlations between $f(B)$ and Δf due to a similar effect of regression to the mean, as discussed in e.g. Berger & Postma (*Genetics*, 2014).

While the dataset we analyzed does not contain replicates from which we can obtain error estimates, we can quantify the extent to which statistical regression to the mean may drive the empirical global epistasis patterns we report. For that, we randomized the pairing between genotypes and fitness values in our landscape (500 randomizations per drug dose), and we analyzed the regressions between $f(B)$ and Δf in these random landscapes — this is now explained in a new Supplementary Materials section. As shown in the new fig. S5, the patterns observed in this randomization controls are largely not compatible with our empirically observed global epistasis patterns, indicating a minor effect of statistical regression to the mean in this particular landscape.

3. I am puzzled by the value provided by the minimal model surrounding Equation 4 and the paragraphs after line 157. Since the fitness measurements of all 2^4 mutations are available, I presume one can simply fit all the fitness landscape parameters (and perhaps apply a threshold to remove the near-zero terms and obtain the simpler model). However, since these parameters are obtained by fitting to the data, they will of course reproduce the empirical patterns. And since Equation 1 is a general mathematical expression for the slope, changing the parameters of the fitness landscape should lead to slopes of different signs. As stated in point 1 above, the change in slopes does not imply that this model leads to linear global epistasis patterns.

As mentioned in our responses to reviewer #1, we now see that the minimal model in our original manuscript was not sufficiently justified. Note that the minimal model was not empirically parametrized; instead, we intended to use it as a tool to provide a geometric interpretation for the situations where different mutation-by-mutation interactions may drive the observed correlations between $f(B)$ and Δf . For that, we used a simple model structure and a set of parameters that qualitatively (but not quantitatively, as the model was not empirically parametrized) reproduced the observed behavior of mutation C59R.

We agree, however, that the utility of the minimal model was not necessarily clear in the original text. We believe that the new results in our revised manuscript provide a more transparent interpretation for how changes in environmental factors may shape global epistasis via the modification of the effective interactions of a mutation. We thus have opted for removing the minimal model from the manuscript altogether.

Reviewer #3

This manuscript takes advantage of previously published data and theory to present an analysis of how global epistasis among four mutations in *P. falciparum* genes is modulated by the environment (antimalarial drugs). The key result is a switch in the direction of global epistasis – from a negative slope to a positive slope – as the drug concentration increases (Fig 1D). The authors employ an existing model to partition these effects into two parts (background fitness and genetic interactions), which when summed together, approximate the real slope data (Fig 3). They then introduce a new, simple model (Equation 4) parametrized with real data, which is claimed to also reproduce the empirical results.

This manuscript is very well-written. I am moderately familiar with the epistasis literature, but it is not my primary area of expertise. I nonetheless found the mathematical explanations clear and easy to follow. I also appreciated the diagrams in Fig 2.

The scientific context and citations provided in the introduction and discussion suggest that the general problem – modulation of global epistasis by environmental variables – is important to the larger goal of predicting adaptation. This paper contributes to understanding this problem by presenting a fully-analyzed, concrete example of the modulation of global epistasis by environmental variables. More specifically, this paper shows that existing epistasis theory holds true in the context of sparse interactions. In this way it appears experimentally similar to Hall et al., 2019 (Evolution), but the mathematical treatment here is more thorough. In general, I think this work makes a notable contribution to the stated problem through its comprehensive treatment of the fitness effects (and interactions) of four mutations in multiple drug environments.

We thank the reviewer for their positive assessment of our work and useful feedback.

Most of the claims in the paper are convincingly supported by the data and models (e.g. Fig 3C and Fig 1D). The main exception, to my eye, was the claim about the accuracy of the simple minimal model (Fig 4). Although it does reproduce the expected slopes, the positions of the points themselves differ substantially between Fig 4E and Fig 4F. Specifically, the example presented in the text (lines 173-179) notes that the filled squares move to the right on the x-axis with increasing drug concentration (Fig 4E). However, in Fig 4F, the filled squares actually move left on the x-axis. Additionally, the points representing ‘both S and N’ are consistently far apart in F despite being nearby in E. Does this mean that mutation I does have a substantial effect that is glossed over in the simplified model? More broadly, what is the benefit of using this simple (but seemingly less accurate) model instead of the more accurate model presented in Fig 3? I find the apparent inaccuracy surprising, since the simple model was parameterized with real data. Perhaps it indicates that this simple model is too simple?

As mentioned in our responses to the other two reviewers, we acknowledge that our original manuscript failed to adequately explain how the minimal model was constructed and its intended purpose. Note that the minimal model was not empirically parametrized. While our goal was for the model to qualitatively reproduce the behavior of the C59R mutation (specifically, the shift from negative to positive global epistasis slope), we did not select parameter values to quantitatively match the observed global epistasis patterns. Our aim was to use the minimal model as a simple tool to provide a geometric interpretation for the “dominance” of different mutation-by-mutation interactions at varying drug doses.

It is evident that both our objective and parameter choices were insufficiently explained in the original manuscript. Following the feedback of the three reviewers, we have opted to remove the minimal model from the paper. We are confident that the new results presented in the current version of the manuscript offer a clearer interpretation for the “dominance” of different gene-by-gene interactions than was previously provided by the minimal model.

Besides additional justification for (or improvements on) this simple model, I have no suggestions for additional experiments or analysis. I find this work to be a thorough and useful analysis of the stated general problem.

We again thank the reviewer for their feedback and hope they will find the new version of the manuscript to be more clear than the original.

Minor comments:

Please name the genes in which the four studied mutations occur.

The four mutations are base substitutions at different positions of the *P. falciparum* gene for the dihydrofolate reductase (DHFR) enzyme. This is now stated in the main text.

Fig 1 and Fig 2: Is this drawing of a bacterium? *P. falciparum* does not have a flagellum nor a single, circular chromosome.

We have updated the drawings to better depict the aspect of a *P. falciparum* cell.

Fig 1D and elsewhere: How is it determined whether the slopes are significantly non-zero? Can significance be indicated in the figure panels, including in the supplement where a different color scheme is used?

As mentioned in our responses to the other two reviewers, we have now extended our analysis to account not only for the slope but also for the coefficient of determination (R^2) of the linear regressions. We show that this R^2 can be written in terms of the coefficient of variation of the distribution of effective interactions for a given mutation (i.e., equation 5 in the new Box 1). While not technically a metric of significance, the R^2 does provide a quantification of the amount of variance explained by the linear model, and thus serves to identify situations where this model may not accurately predict the fitness effect of a mutation.

We have also updated the color scheme in the supplementary figures to match that used in the main text.

Fig 3C predicts that the slope of the global epistasis line will be positive at drug concentration 100. In Fig 1D, however, the slope is zero. Is there any explanation for this discrepancy?

As noted by the reviewer, we often observe small deviations between the empirically observed slopes and the values estimated using equations 1-5 (see the new Fig. 4). The equations in our original manuscript (now summarized and expanded in the new Box 1) are low-order approximations which only partially account for higher-order (i.e., higher than pairwise) epistasis. This necessarily indicates that any discrepancy must be due to a substantial effect of higher-order epistasis. We have attempted to make this more clear in the main text, and also added a new Supplementary Materials section explaining how our equations are derived, explicitly showing the low-order approximation.

Please define “microscopic genetic interactions.” I see that several citations are provided after this term, but I still find it confusing, since “microscopic” has a much more common meaning in biology that is unrelated to epistasis.

This is a fair point. In the original manuscript (and often in the global epistasis literature), the term “microscopic epistasis” is used in contrast to “global” epistasis, referring to the fine-grained interaction between pairs of loci rather than the global interaction emerging in the form of a relationship between $f(B)$ and Δf . We see how this terminology might be confusing. In the new manuscript we have avoided the term “microscopic interactions,” using the terms “pairwise” or “fine-grained interactions” where appropriate.

Reviewers' Comments:

Reviewer #1:

Remarks to the Author:

Kudos to the authors for this thoughtful revision (rewrite!!). I find it much clearer and more insightful. I am eager to see how this impacts the field - especially the eostasis trajectories.

I have no further comments.

Jake Scott

Reviewer #2:

Remarks to the Author:

I thank the authors for considering comments from the previous round of reviews. Considering the revisions made by the authors, it is still difficult not to conclude that this work is incremental w.r.t recent work in the field. To revisit recent activity in the field:

- 1) Gene-by-gene interactions lead to global epistasis, as shown by refs 43 and 47.
- 2) This is validated by predictions tested on data from refs 12, 14 and others. Importantly, ref. 14 contains data from a 10-site fitness landscape across six environments. This analysis points to how global epistasis trends are modulated by environmental variation.
- 3) A nice re-interpretation/review of refs 43, 47 and others has been published by the authors of the current paper in <https://royalsocietypublishing.org/doi/full/10.1098/rstb.2022.0053> (arxiv version is ref 44). This paper develops the theory presented in Box 1 of the current paper. Indeed, the current paper overall makes the same point as Figure 3 of the above paper in Phil. Trans. R. Soc. B.

Put together, these three works show that global epistasis trends will arise with appropriate structure (or lack thereof) in gene-by-gene interaction terms of the fitness landscape.

To re-iterate key points from my previous report, I have two fundamental concerns with the current paper:

- i) That the analysis of the 4-site landscape in this paper does not contain new insights w.r.t recent work cited above, and
- ii) As noted by the others, most of the linear fits are not up to par (more than 80% of them have an $R^2 < 0.5$). This is likely because of the small size of the dataset. The authors state in their rebuttal that linearity is not a point they wanted to make in the manuscript. This is quite puzzling, however, as this paper aims to determine the basis of global epistasis trends, which are defined as linear relationships between fitness effects and background fitness. I am uncertain how to interpret most of the data presented here as 'global epistasis trends' if most of this data is not or cannot be determined to be linear.

Reviewer #3:

Remarks to the Author:

In their revision, Diaz-Colunga et al. have made substantial changes to their manuscript. I do applaud the efforts that the authors have made to address comments from the other reviewers in a positive and professional way. Unfortunately, though, I am unable to be as positive about the revised version as I was about the original version.

One issue is that, with the sizable addition of novel theory, the paper has lost a great deal of readability and accessibility to biologists. I did appreciate the addition of the specific definitions of

'strength of epistasis' and 'the degree to which epistasis is global'; and I think that Figure 1C and 1D illustrate these quantities well. In contrast, the paper is missing a thorough justification or explanation of the "effective interaction" term. An equation is presented in Box 1, but where did it come from? What is the logic behind it? Why is (epistasis between mutations) divided by (fitness of one of those mutations) called an "effective interaction" at all? Perhaps this is carefully walked through in the cited publications, but in this revised manuscript, the reasoning behind the math is virtually absent.

The authors claim that the "effective interaction" term predicts global epistasis, but since this is not really true (Figure 3F), it does not help me understand what an "effective interaction" is. How impressive is it that "effective interactions" predict the strength of epistasis (Figure 3D) when the definition of the term itself includes the strength of epistasis? Finally, when all the interesting variation in R^2 (Figure 2E) is averaged to a $y=1$ line in Figure 3F, what exactly is "effective interaction" really telling us about R^2 ?

For the 'weighted' term 'w', I am concerned about its dependence on the number of mutations being considered. Although it seems that 'w' will always be quite small, since its numerator is just one number (average) versus the multiple averages summed together in the denominator; it also seems that the more genotypes in question, the smaller and smaller 'w' will become. This seems to mean that global epistasis will appear smaller and smaller as more and more alleles are considered. Is this intended to be true? Given the minimal explanation of the logic of 'w', I am also concerned that the reference for the definition of 'w' is a review by the authors (Ref. 44) instead of a theory paper.

Finally, I think the authors need to address the fact that they are trying to apply their newly-developed, general epistasis theory to one specific, and peculiar, biological example. In contrast to Bakerlee et al., which examined epistasis between mutations in 10 unlinked genes, this work considers four mutations in a single gene that are all known to have the same phenotypic effect (i.e., drug resistance). The theory in the manuscript relies quite heavily on averaging fitness effects against all possible backgrounds with and without these mutations. But of course, the fitness effects of these mutations on these backgrounds will be bimodal: in the presence of a drug, one resistance mutation on a non-resistant background will be very fit; while the same resistance mutation on a background that is already resistant will not add any, or very little, fitness. Similarly, the degree of decline in fitness in the absence of drug will depend on whether another low-fitness mutation is already present or not. Averaging this all together, instead of treating it as two phenotypically distinct scenarios, does not seem very biologically informed. In Figure 3A, for example, the authors seem to claim that two of these drug resistance mutations do not actually confer drug resistance. Could it be that fitness is positive on truly WT backgrounds, and negative on backgrounds that already have resistance mutations?

Minor comments:

"Allele" is used to mean "genotype" in the Figure 1A caption, Line 48, and throughout the manuscript. Since "alleles" refer to variations at single sites, I don't think it's the correct term when referring to combination of alleles across four sites.

Figure 1A caption says there are 16 "alleles," but the text says 15.

While a gametocyte is a better illustration than a bacterium, it still is inaccurate, assuming the experiments were conducted in vitro. Why not show the parasite as a ring, trophozoite, or shizont?

Box 1 figure: calling the alleles by their first letter (e.g. I for I164L) is nonsensical because, at least the field of malaria drug resistance, the first letter (I) is the wild-type and the second letter (L) is the mutation. I also found the network figure hard to parse because the focal genotypes hardly stand out from the background. It would be clearer if the background genotypes were transparent.

Box 1 still says "microscopic interactions". It is fine if this is the normal term, just please define it at first use.

Reviewer #1

Kudos to the authors for this thoughtful revision (rewrite!!). I find it much clearer and more insightful. I am eager to see how this impacts the field - especially the epistasis trajectories.

I have no further comments.

Jake Scott

We again thank the reviewer for his very helpful feedback and positive assessment of our work.

Reviewer #2

I thank the authors for considering comments from the previous round of reviews. Considering the revisions made by the authors, it is still difficult not to conclude that this work is incremental w.r.t recent work in the field. To revisit recent activity in the field:

1) Gene-by-gene interactions lead to global epistasis, as shown by refs 43 and 47.

2) This is validated by predictions tested on data from refs 12, 14 and others. Importantly, ref. 14 contains data from a 10-site fitness landscape across six environments. This analysis points to how global epistasis trends are modulated by environmental variation.

3) A nice re-interpretation/review of refs 43, 47 and others has been published by the authors of the current paper in [URL] (arxiv version is ref 44). This paper develops the theory presented in Box 1 of the current paper. Indeed, the current paper overall makes the same point as Figure 3 of the above paper in *Phil. Trans. R. Soc. B*.

Put together, these three works show that global epistasis trends will arise with appropriate structure (or lack thereof) in gene-by-gene interaction terms of the fitness landscape.

To re-iterate key points from my previous report, I have two fundamental concerns with the current paper:

i) That the analysis of the 4-site landscape in this paper does not contain new insights w.r.t recent work cited above

We would like to thank the reviewer for their careful reading of our manuscript and their thoughtful comments. We have tried our best to address them in the revised manuscript. With regard to the degree of incrementality of our findings, we of course appreciate that this is a continuum, and where exactly a particular manuscript lies on this continuum is ultimately a judgment call. We do believe our paper makes meaningful points that are important for the scientific literature; just for the sake of laying our case as clearly as possible, we list these below. We have also done our best to clarify in the revised manuscript what we believe the main contributions of the paper are. As we say, however, this is ultimately a judgment call and we respect and accept the opinion of the reviewer on this matter.

The main contributions of our paper are the following:

- Our study analyzes a fitness landscape of intrinsic biological significance, consisting of four mutations that have been found to be associated with drug resistance in malaria throughout multiple geographical locations world wide (e.g., Tahar & Basco 2006, Ahmed et al. 2006, Heidari et al. 2007, Gebru-Woldearegai et al. 2005). In our manuscript, we study the fitness landscape made up by these four mutations in different environment consisting of varying concentrations of two different antimalarial drugs, both used in clinical contexts. We believe that examining how microscopic epistasis gives rise to global patterns in biomedically relevant fitness landscapes is an important step if we wish to determine how significant global epistasis is beyond model organisms in a laboratory setting.
- While focusing on a particular, low-dimensional fitness landscape of course has limitations, it has the important advantage of allowing us to quantify every possible interaction between all pairs of mutations, which is more challenging to do in larger combinatorial spaces. In this paper, this allows us to first characterize the effect of drug dosage on microscopic epistasis and on the average fitness effect of the four drug resistance mutations. From here, we can discriminate if the changes in global epistasis at increasing drug concentrations are due to the former, the latter, or, as is the case in our empirical drug-resistance landscapes, to more or less equal contributions from both.

- We show that a low-order approximation to microscopic epistasis can serve to explain the strength and shape of global epistasis across environments. In previous work (ref. 44, now ref. 49 as the reviewer notes) we showed this was true for the slope of the global epistasis regression, but the observation that the variance ratio $\text{var } \Delta f / \text{var } f(B)$ and the R^2 of the regression can be related to the same distribution of “effective interactions” is a result of the current paper. We do not believe this has been examined previously. It is also important to note that ref. 43 (now 48) originally assumed widespread epistasis, but here we provide a direct empirical example of the same quantitative framework explaining global epistasis patterns when a single interaction between mutations dominates.

We have emphasized these aspects throughout the revised version of the manuscript, and have attempted to make our conclusions more specific to our particular fitness landscape. As we state above, we believe this represents an important contribution to the field, but we respect the reviewer’s judgment and opinion about this matter. Below, we address the more technical comments made by the reviewer in their last round of revision. We hope that our response will clarify the issues that were raised.

ii) As noted by the others, most of the linear fits are not up to par (more than 80% of them have an $R^2 < 0.5$). This is likely because of the small size of the dataset. The authors state in their rebuttal that linearity is not a point they wanted to make in the manuscript. This is quite puzzling, however, as this paper aims to determine the basis of global epistasis trends, which are defined as linear relationships between fitness effects and background fitness. I am uncertain how to interpret most of the data presented here as ‘global epistasis trends’ if most of this data is not or cannot be determined to be linear.

Regarding the question of linearity, we think there might have been a miscommunication with respect to our understanding of the issue and that of the reviewer. Thus, we would like to first clarify our argument that “linearity was not a main point we intended to make in our manuscript.” In our view, epistasis can have a *global* component when the fitness effect of a mutation can be linked to the fitness of the genetic background, and thus can be predicted with some degree of accuracy from the background fitness (that is, without prior information on which other mutations may be present or absent from the background). There is, however, a fundamental question of when one might expect the relationship between Δf and $f(B)$ to be truly linear. We think the reviewer might be referring to two potential situations where this may not be the case.

First, it has been described that relationships between Δf and $f(B)$ can emerge as a result of epistatic interactions between mutations. The shape of these relationships depends on the structure of such interactions. This idea is perhaps best articulated in a recent review by Johnson et al. (*BMC Biology*, 2023; now ref. 15 in our manuscript). We reproduce one of the figures of that paper below.

Fig. 1. Extracted from Johnston et al., *BMC Biology* (2023).

The figure above illustrates how, if a mutation does not interact with any other loci, its fitness effect remains roughly constant across all genetic backgrounds (left panel). If the mutation interacts with only one other genomic site, the fitness effect “splits” into two, and becomes different depending on whether the genetic background carries or does not carry that second mutation (middle panel). Naturally, in this scenario the relationship between Δf and $f(B)$ will not be linear, and would instead be best described by a piece-wise function such as

$$\Delta f = \begin{cases} A & \text{if mut. } j \text{ in background} \\ B & \text{if mut. } j \text{ not in background} \end{cases}$$

Still, one could fit a linear model linking Δf to $f(B)$ (instead of linking Δf to the presence/absence of mutation j as in the expression above). Importantly, such a linear model would allow us to predict the fitness effect from only the background fitness with *some* accuracy (even if not perfectly). Of course, there would be inevitable deviations from the linear model, which in our framework we interpret as a lower “degree to which epistasis is global.”

In the rightmost panel Johnston et al. illustrate the situation where the focal mutation engages in widespread interactions with other loci, in which case the collective effect of all interactions leads to the emergence of an apparently linear correlation between Δf and $f(B)$. It is important to note, however, that in this latter case deviations from the linear fit will generally be observed *even in the absence of measurement error*. In fact, Reddy & Desai (*eLife*, 2021) provide a quantitative description of how microscopic epistasis dictates the structure of these deviations. Thus, despite the rightmost panel illustrating a seemingly “more linear” trend than the middle panel, it is not necessarily the case that this relationship can be interpreted as “truly linear with noise.” The linear fit might work better at estimating Δf in the rightmost panel than in the middle panel, but that does not mean that there exists a true, fundamental linear relationship between Δf and $f(B)$ in either of the two cases.

It is also important to note that the distinction between these two scenarios is often quantitative, not qualitative. In the middle panel, it need not be the case that there exists a *single* pairwise interaction between loci; instead, this type of pattern would emerge if one particular interaction was substantially stronger than all the others. This can be rationalized in terms of the distribution of interactions that we discuss in our paper. In our data, we indeed see this type of “piece-wise” pattern (e.g., in the last panel of Fig. 2B in the manuscript) when the distribution of interactions is very skewed due to a single, strong interaction (that of mutation C59R with mutation S108N at high drug doses, as shown in Fig. 3C in the manuscript). This observation also highlights the utility of analyzing the low-dimensional landscape in our paper: while the number of data points is more limited than in other studies, the small size of this dataset allows us to directly quantify almost every interaction between mutations across genetic backgrounds and drug doses, which is more challenging in higher-dimensional datasets.

Alternatively, the reviewer’s concern may have to do with the question of whether the relationship between Δf and $f(B)$ is truly linear in our data — in other words, whether we should see a perfect linear correlation if measurement error was eliminated. Previous work has shown that true linearity will emerge only when there exists a latent variable on which mutations act additively, with fitness being an exponential-like transformation of this variable (e.g., Otwinowski et al. 2018). For illustrative purposes, let us consider an additive variable λ such that for a given genotype G :

$$\lambda(G) = \sum_i \lambda_i x_i \quad (1)$$

where $x_i = 0, 1$ denotes the presence or absence of mutation i , and λ_i is the contribution of each mutation to the latent variable. If the fitness of G is given by an exponential transformation of the generic form

$$f(G) = 1 - \exp(-\lambda(G)) \quad (2)$$

it is true that Δf and $f(B)$ will be linearly correlated — as noted by the reviewer, we discuss this in detail in a recent review (ref. 44, now ref. 49). In the illustration below, we consider 6 mutations and randomly sample λ_i from a normal distribution, and we plot the fitness effect of one of the mutations against the background fitness:

Fig. 2

Note, however, that if the transformation on λ is non-exponential, the relationship between Δf and $f(B)$ will generally not be linear. As an example, if we consider:

$$f(G) = \frac{\lambda^2(G)}{1 + \lambda^2(G)} \quad (3)$$

we obtain the plot below:

Fig. 3

The true relationship between Δf and $f(B)$ is now non-linear, although of course one can always fit a linear model to the data (blue line in rightmost panel) which may have some predictive power. In practice, identifying non-linear relationships between Δf and $f(B)$ can be challenging if there is additional noise (be it due to underlying biological processes or simply measurement error). For example, if we consider:

$$f(G) = \frac{\lambda^2(G)}{1 + \lambda^2(G)} + \underbrace{\sigma(G)}_{\text{random noise}} \quad (4)$$

we find that the latter plot turns into:

Fig. 4

From this last plot, it is naturally difficult to determine what the true functional form is of the relationship between Δf and $f(B)$. Note that there are recent methodologies which aim to identify latent additive variables (what we denote as λ) in order to reconstruct genotype-phenotype maps (e.g., Otwinowski et al. 2018, Tonner et al. 2022) — but such latent variables might not always exist, or they might not carry a significant biological meaning. With the dataset we are analyzing in our paper, we do not think it would be justified to claim that the relationships we observe between Δf and $f(B)$ are truly linear (as in Fig. 2 in this report) or otherwise (as in Figs. 3 and 4 in this report): this is why we argued that linearity was not a main point we intended to make.

In summary, the relationship between Δf and $f(B)$ will be truly linear if and only if fitness is an exponential-like transformation of a latent additive variable, which may often not be the case. From the perspective of our paper, we take a more parsimonious approach and consider that epistasis can have a *global* component if the fitness effect of the mutation can be predicted (to some accuracy) from a linear model linking it to the background fitness — even if the true relationship between the two magnitudes may not be truly linear. The linear models are thus not intended to convey that the relationships between Δf and $f(B)$ must be truly linear in our data, which is what we meant when we said that “linearity was not a main point we intended to make in our manuscript.”

The reviewer is absolutely correct that the linear models do not always work well: in many cases, the prediction of the model is not significantly better than what one could have predicted simply from a mutation’s average fitness effect, resulting in low R^2 values. Again for the sake of parsimony, we interpret these situations as cases where epistasis is largely “non-global” — regardless of whether this happens because the true relationship between Δf and $f(B)$ is non-linear (as in Fig. 3 of this report) or because the variation in Δf may be better captured by a “piece-wise” model (as in Fig. 1 of this report, middle panel).

Still, note that most mutations (3 out of 4 for pyrimethamine and all four for cycloguanil) reach high R^2 (> 0.5) at *some* (but not all) drug doses. Importantly, the conceptual framework in our paper can also explain the situations where the correlation between Δf and $f(B)$ is weak: these would correspond to regimes where the focal mutation interacts only weakly with other loci, i.e., the distribution of effective interactions is narrow and centered at zero (bottom left corner of Fig. 1D in the manuscript, this situation is seen empirically, for instance, for mutation C59R at 10^{-1} μM of cycloguanil), or interacts strongly with other loci but with positive and negative interactions “balancing out,” i.e., the distribution of effective interactions is wide and centered at zero (bottom right corner of Fig. 1D in the manuscript, seen empirically, for instance, for mutation S108N at 10^2 μM of cycloguanil). The observation that drug concentration can vary the strength of the relationship between Δf and $f(B)$ (the “degree of global epistasis”), and that this effect can be traced back to the same distribution of interactions which explains the slope of the regressions, is precisely one of the results of our paper — we have now tried to clarify this in the manuscript.

Reviewer #3

In their revision, Diaz-Colunga et al. have made substantial changes to their manuscript. I do applaud the efforts that the authors have made to address comments from the other reviewers in a positive and professional way. Unfortunately, though, I am unable to be as positive about the revised version as I was about the original version.

One issue is that, with the sizable addition of novel theory, the paper has lost a great deal of readability and accessibility to biologists. I did appreciate the addition of the specific definitions of ‘strength of epistasis’ and ‘the degree to which epistasis is global’; and I think that Figure 1C and 1D illustrate these quantities well. In contrast, the paper is missing a thorough justification or explanation of the “effective interaction” term. An equation is presented in Box 1, but where did it come from? What is the logic behind it? Why is (epistasis between mutations) divided by (fitness of one of those mutations) called an “effective interaction” at all? Perhaps this is carefully walked through in the cited publications, but in this revised manuscript, the reasoning behind the math is virtually absent.

We thank the reviewer for their thorough assessment of our revised manuscript. We do see how the original manuscript, focusing on the slope of only a single mutation, was more easily accessible to a broad audience. At the same time, we agree with reviewer #1 in that the revised version of the manuscript provides new valuable insights with respect to the original one. In this second revision, we have attempted to maintain these insights while providing more clear interpretations for the mathematical expressions presented.

With respect to the equations in Box 1, a more detailed derivation can be found in the Supplementary Material. This derivation builds on previous theoretical work (refs. 43 & 44, now 48 & 49): in short, the expressions in Box 1 result from considering only the lowest-order terms in the expansions for the variance of Δf , the variance of $f(B)$, and the covariance between Δf and $f(B)$. While we think that fully reproducing the original analyses in refs. 43 and 44 (now 48 & 49) would not be appropriate here, we have attempted to make the low-order approximation more clear in Box 1 itself and in the Supplementary Material.

The authors claim that the “effective interaction” term predicts global epistasis, but since this is not really true (Figure 3F), it does not help me understand what an “effective interaction” is. How impressive is it that “effective interactions” predict the strength of epistasis (Figure 3D) when the definition of the term itself includes the strength of epistasis? Finally, when all the interesting variation in R^2 (Figure 2E) is averaged to a $y=1$ line in Figure 3F, what exactly is “effective interaction” really telling us about R^2 ?

Deviations from the empirically observed variance ratio, R^2 , and slope with respect to the values provided by equations 1-5 can be attributed to the effect of higher-order interactions (HOIs) on these magnitudes: as explained above, the equations in Box 1 result from truncating the expressions for the variances and covariance of Δf and $f(B)$ at the lowest-order terms. We have attempted to state this approximation more clearly in the mathematical exposition of the equations. We also now provide a hopefully more intuitive interpretation for what we call an “effective interaction.” In short, since global epistasis patterns emerge as relationships between Δf

(y-axis) and $f(B)$ (x-axis), effective interactions are the composition of two contributions to these relationships: a “vertical” contribution to the variation in Δf (captured by the $\langle \epsilon_{ij} \rangle$ terms), and a “horizontal” contribution to the variation in $f(B)$ (captured by the $\langle \Delta f_j \rangle$ terms). In the end, what we call an “effective interaction” (β_{ij}) and its corresponding “weight” (ω_{ij}) are just the result of conveniently arranging these terms so that the variance ratio, R^2 , and slope can be expressed in terms of different moments of a same distribution.

The reviewer is of course right in that a mutation which engages in strong interactions with other loci (large ϵ_{ij}) can be naturally expected to exhibit large variation in its fitness effect across genetic backgrounds. However, we believe that our results go beyond this qualitative (and perhaps “not impressive”) observation: by defining effective interactions, we are able to *quantitatively* link the strength of fine-grained epistatic interactions (what is often referred to as “microscopic” epistasis) with the three aspects of epistasis we consider (overall strength of epistasis, degree of global epistasis, and shape of global epistasis) via the properties (moments) of a *same* distribution. Note that this would not be the case if we directly analyzed the “raw” distributions of fitness effects and/or the distributions of epistatic effects. Note also that a mutation i may engage in similarly strong interactions with two other mutations j and k (similarly large ϵ_{ij} and ϵ_{ik}), but the contribution of these interactions will be substantially different if j and k have different fitness effects (e.g., $\Delta f_j \gg \Delta f_k$). We have attempted to make this more clear throughout the manuscript.

It is important to note that the effect of HOIs is partially (but only partially) accounted for in equations 1-5: for instance, the fact that the interaction ϵ_{ij} between mutations i and j appears averaged across all genetic backgrounds (as $\langle \epsilon_{ij} \rangle$) means that in general we may expect said interaction to change in magnitude depending on the presence/absence of additional background mutations — which would by definition indicate the existence of HOIs. If we did not neglect these HOIs, equations 1-5 would contain additional terms depending on $\langle \epsilon_{ijk} \rangle$ (defined as the average deviation in fitness of the triple mutant with respect to the additive expectation from each of the three single mutants), $\langle \epsilon_{ijkl} \rangle$, etc. The larger the magnitude of such terms, the more substantial the deviation we will observe between the empirically measured variance ratio, R^2 , and slope and the values given by equations 1-5.

We agree with the reviewer that the previous version of our manuscript did not explain the origin of the observed deviations between the measured variance ratio, R^2 , and slope with respect to the estimations provided by equations 1-5. There are, in fact, some interesting observations in this regard: for instance, there seems to be a better agreement between the estimated and measured R^2 for mutation C59R at low pyrimethamine doses than at high concentrations (Fig. 4D, red dots) — this indicates that the effect of HOIs for this particular mutation becomes more prominent as the dose of pyrimethamine increases. Interestingly, while cycloguanil modulates the global epistasis pattern of C59R in a similar fashion to pyrimethamine (both drugs have similar mechanisms of action), the agreement between the observed and estimated R^2 seems to be better for cycloguanil (Fig. S4D, red dots), indicating a smaller effect of HOIs. We now discuss these observations explicitly throughout the manuscript.

For the ‘weighted’ term ‘w’, I am concerned about its dependence on the number of mutations being considered. Although it seems that ‘w’ will always be quite small, since its numerator is just one number (average) versus the multiple averages summed together in the denominator; it also seems that the more genotypes in question, the smaller and smaller ‘w’ will become. This seems to mean that global epistasis will appear smaller and smaller as more and more alleles are considered. Is this intended to be true? Given the minimal explanation of the logic of ‘w’, I am also concerned that the reference for the definition of ‘w’ is a review by the authors (Ref. 44) instead of a theory paper.

The reviewer is correct that the weights ω_{ij} may be reasonably expected to become smaller as the dimension of the landscape is increased. However, this need not be the case in general.

First, considering additional mutations increases the number of potential genetic backgrounds over which the terms $\langle \Delta f_j \rangle$ are averaged. In equation 3,

$$\omega_{ij} \equiv \frac{\langle \Delta f_j \rangle^2}{\sum_{k \neq i} \langle \Delta f_k \rangle^2}$$

while the number of terms in the denominator sum increases, the values of all $\langle \Delta f_j \rangle$ can also change (either increase, decrease, or remain roughly the same); and thus the final value for the weight may increase or decrease. Still, we definitely agree with the reviewer’s expectation that, in general, adding mutations (and therefore genotypes) will make most ω_{ij} smaller.

Yet, importantly, this does not necessarily mean that global epistasis will become weaker. What it indicates is that the *individual contribution* of any one background mutation to the emergent pattern of global epistasis will decrease. But, since there are also more mutations contributing to the global epistasis pattern, it is entirely possible that epistasis remains strongly “global,” or even becomes stronger.

Perhaps this idea can be best illustrated with a simple example. Let us consider a mutation (mut. 1) that engages in epistatic interactions with, say, three other mutations (mut. 2, 3, and 4). For simplicity, consider that all $\epsilon_{ij} = 1$ and $\Delta f_j = 1$ (in arbitrary units) for all mutations. We therefore get all effective interactions $\beta_{ij} = 1$ and all weights $\omega_{ij} = 1/3$. If we use equation 5 to estimate the R^2 (the “degree of global epistasis”) for mutation 1, we obtain:

$$R^2 \approx \frac{(\omega_{12}\beta_{12} + \omega_{13}\beta_{13} + \omega_{14}\beta_{14})^2}{\omega_{12}\beta_{12}^2 + \omega_{13}\beta_{13}^2 + \omega_{14}\beta_{14}^2} = \frac{(1/3 \cdot 1 + 1/3 \cdot 1 + 1/3 \cdot 1)^2}{1/3 \cdot 1^2 + 1/3 \cdot 1^2 + 1/3 \cdot 1^2} = 1$$

Indicating that, in this simple example, epistasis would be “fully global” ($R^2 = 1$).

If we now consider an additional mutation (mut. 5), we notice that each individual weight indeed becomes smaller ($\omega_{ij} = 1/4$ instead of $1/3$). But, at the same time, there is now an additional term capturing the contribution of the interaction between muts. 1 and 5 to the global epistasis pattern (highlighted in red in the expressions below):

$$R^2 \approx \frac{(\omega_{12}\beta_{12} + \omega_{13}\beta_{13} + \omega_{14}\beta_{14} + \omega_{15}\beta_{15})^2}{\omega_{12}\beta_{12}^2 + \omega_{13}\beta_{13}^2 + \omega_{14}\beta_{14}^2 + \omega_{15}\beta_{15}^2} = \frac{(1/4 \cdot 1 + 1/4 \cdot 1 + 1/4 \cdot 1 + 1/4 \cdot 1)^2}{1/4 \cdot 1^2 + 1/4 \cdot 1^2 + 1/4 \cdot 1^2 + 1/4 \cdot 1^2} = 1$$

and so the strength of global epistasis remains the same ($R^2 = 1$). Note that, for instance, if we had taken the interaction between muts. 1 and 2 to be negative ($\epsilon_{12} = -1$, blue terms in expressions below), we would first have:

$$R^2 \approx \frac{(1/3 \cdot (-1) + 1/3 \cdot 1 + 1/3 \cdot 1)^2}{1/3 \cdot (-1)^2 + 1/3 \cdot 1^2 + 1/3 \cdot 1^2} = 0.11$$

But, after adding the fifth mutation (with positive epistasis between muts. 1 and 5):

$$R^2 \approx \frac{(1/4 \cdot (-1) + 1/4 \cdot 1 + 1/4 \cdot 1 + 1/4 \cdot 1)^2}{1/4 \cdot (-1)^2 + 1/4 \cdot 1^2 + 1/4 \cdot 1^2 + 1/4 \cdot 1^2} = 0.25$$

the strength of global epistasis in fact *increases* in this case, even though each of the individual contributions are smaller.

Naturally, this is only an example and it is entirely possible that the strength of global epistasis does decrease when considering additional mutations (for instance, if mutations 2 to 4 engage in positive epistasis with mutation 1, but epistasis between mutations 1 and 5 is negative). In the end, the effect that additional mutations will have on the strength of global epistasis will depend on the structure of the interactions between all mutations with the new one.

Importantly, we believe that the conceptual framework in our paper can be of help when trying to rationalize these effects. Since the R^2 depends on the coefficient of variation of the distribution of effective interactions, we can intuitively reason that a new mutation may weaken global epistasis if it “widens” the distribution of effective interactions and vice-versa. In future work, it could be interesting to study when we might expect to see each of these behaviors (for instance, considering fitness landscapes of increasing size by adding mutations in increasingly distal regions of a gene). While this falls out of the scope of our current paper, we agree with the reviewer that these are important questions, which we now suggest as a potential directions for future research in the revised manuscript.

Finally, I think the authors need to address the fact that they are trying to apply their newly-developed, general epistasis theory to one specific, and peculiar, biological example. In contrast to Bakerlee et al., which examined epistasis between mutations in 10 unlinked genes, this work considers four mutations in a single gene that are all known to have the same phenotypic effect (i.e., drug resistance). The theory in the manuscript relies quite heavily on averaging fitness effects against all possible backgrounds with and without these mutations. But of course, the fitness effects of these mutations on these backgrounds will be bimodal: in the presence of a drug, one resistance mutation on a non-resistant background will be very fit; while the same resistance mutation on a background that is already resistant will not add any, or very little, fitness.

We completely agree with the reviewer that our results are specific to the fitness landscape we are analyzing, and that the mechanisms governing global epistasis in this study need not apply to other biological settings. Perhaps following on our response to the reviewer’s previous comment, we also agree that studying how different structures of interactions between loci (corresponding to a single or multiple genes, or even a same or different pathways) will be important in future work. In the revised manuscript, we further focus our conclusions on the specific dataset we analyze, particularly in the Discussion section. We also more explicitly justify our choice of landscape: as mentioned in our response to reviewer #2, the dataset we study is of biological relevance for the resistance of a human pathogen to antiparasitic drugs. In addition, the low dimensionality of this dataset, with a quasi-complete set of genotypes, allowed us to quantify every potential interaction between loci across genetic backgrounds and drug doses — which is more challenging to do in larger combinatorial spaces.

Similarly, the degree of decline in fitness in the absence of drug will depend on whether another low-fitness mutation is already present or not. Averaging this all together, instead of treating it as two phenotypically distinct scenarios, does not seem very biologically informed. In Figure 3A, for example, the authors seem to claim that two of these drug resistance mutations do not actually confer drug resistance. Could it be that fitness is positive on truly WT backgrounds, and negative on backgrounds that already have resistance mutations?

Figure 3A in particular highlights that a specific mutation may only confer resistance when combined with others, and/or in specific environments (at certain drug doses but not others). The reviewer is correct that all four mutations confer some level of resistance (positive fitness effects) at high doses, but are deleterious at low doses (negative fitness effects). In the figure below, we show the fitness effect of each of the four mutations on the wild-type genotype:

Interestingly, epistasis appears to be playing a nuanced role here. If there was no epistasis, the above fitness effects of all four mutations would combine additively. If this was the case, we would expect the fitness of the quadruple mutant to be given by:

$$f(\text{quad. mutant}) = f(\text{w.t.}) + \Delta f_{\text{C59R}}(\text{w.t.}) + \Delta f_{\text{I164L}}(\text{w.t.}) + \Delta f_{\text{N511}}(\text{w.t.}) + \Delta f_{\text{S108N}}(\text{w.t.})$$

In the figure below, we plot the actual measured fitness of the quadruple mutant as well as the expectation if there was no epistasis, given by the equation above.

Consistent with the reviewer’s reasoning, epistasis makes the fitness of the quadruple mutant lower than the additive expectation at high drug doses. We entirely agree with the reviewer that redundancy is likely an important mechanism which can explain why the combination of two or more mutations, each of which is beneficial independently, may lead to a decline in fitness. This has been often described in the context of antimicrobial resistance (e.g., in refs. 46 & 47 in the revised manuscript), and we now mention it in our paper.

But it is also important to note that, at low drug doses, epistasis makes it so the fitness of the quadruple mutant is *higher* than the additive expectation. Put simply, this seems to indicate that the interaction structure of these particular four mutations makes it so their combination is “not so fit” in the presence of the drug, but also “not so deleterious” in its absence. We believe this highlights the importance of studying epistasis in this particular fitness landscape, as it might help explain why several studies have found these mutations ubiquitously across many different geographical locations (e.g., Tahar & Basco 2006, Ahmed et al. 2006, Heidari et al. 2007, Gebru-Woldearegai et al. 2005).

The above figures have been added to the manuscript as fig. S1 and are now discussed in the main text.

Minor comments:

“Allele” is used to mean “genotype” in the Figure 1A caption, Line 48, and throughout the manuscript. Since “alleles” refer to variations at single sites, I don’t think it’s the correct term when referring to combination of alleles across four sites.

We have changed “alleles” for “genotypes” where appropriate.

Figure 1A caption says there are 16 “alleles,” but the text says 15.

We have corrected the legend of Fig. 1.

While a gametocyte is a better illustration than a bacterium, it still is inaccurate, assuming the experiments were conducted in vitro. Why not show the parasite as a ring, trophozoite, or shizont?

We have updated the drawing in Figs. 1, 2 and 3.

Box 1 figure: calling the alleles by their first letter (e.g. I for I164L) is nonsensical because, at least the field of malaria drug resistance, the first letter (I) is the wild-type and the second letter (L) is the mutation. I also found the network figure hard to parse because the focal genotypes hardly stand out from the background. It would be clearer if the background genotypes were transparent.

We now use numbers (1 to 4) instead of letters to identify mutations. We have also further highlighted the focal genotypes.

Box 1 still says “microscopic interactions”. It is fine if this is the normal term, just please define it at first use.

We had missed the instance in Box 1 where we still referred to “microscopic interactions.” This is now corrected.

Reviewers' Comments:

Reviewer #2:

Remarks to the Author:

I very much appreciate the authors' detailed response to my comments. I am still not enthusiastic about the results here, for the reasons expanded upon in my previous two reports -- namely, that (1) in my view, the analysis does not provide new insights relative to established literature, and (2) the global epistasis patterns presented here reflect predictable trends in the genotype-phenotype map.

As the authors note in their rebuttal, the former point is a matter of personal judgment and I leave it to the editors to establish the value of the work given the literature cited in the paper and highlighted in my previous report.

Regarding the latter point -- I thank the reviewers for their extensive rebuttal. Global epistasis indeed refers to the phenomenon where fitness effects of mutations can be well-predicted based on background fitness. In my view, the data shown in Figures S2 and S3 (where the data for all the mutations is presented) does not show such predictable trends in most cases, which is also reflected in the low R^2 values for the linear fits. I agree with the authors that global epistasis trends may not necessarily be linear, however, it is hard to glean any trend, linear or non-linear, from visualizing the raw data for most of the plots. Again, I defer to the editor to evaluate this judgment.

Reviewer #3:

Remarks to the Author:

In general, exploring how epistasis changes with environmental conditions is a worthy goal. I appreciate the additional space given to the important idiosyncrasies of this particular system in the second revision (Lines 50-62). The paper (and responses to reviewers) are very clearly written, and since the revised paper itself does not exactly respond to several of the reviewer criticisms, I will trust that the authors are happy with it as it is. The work is quite professionally presented, and I'm sure it will be interesting to others working in epistatic theory.

RESPONSE TO REVIEWER COMMENTS (in bold text).

Reviewer #2 (Remarks to the Author):

I very much appreciate the authors' detailed response to my comments. I am still not enthusiastic about the results here, for the reasons expanded upon in my previous two reports -- namely, that (1) in my view, the analysis does not provide new insights relative to established literature, and (2) the global epistasis patterns presented here reflect predictable trends in the genotype-phenotype map.

As the authors note in their rebuttal, the former point is a matter of personal judgment and I leave it to the editors to establish the value of the work given the literature cited in the paper and highlighted in my previous report.

Regarding the latter point -- I thank the reviewers for their extensive rebuttal. Global epistasis indeed refers to the phenomenon where fitness effects of mutations can be well-predicted based on background fitness. In my view, the data shown in Figures S2 and S3 (where the data for all the mutations is presented) does not show such predictable trends in most cases, which is also reflected in the low R^2 values for the linear fits. I agree with the authors that global epistasis trends may not necessarily be linear, however, it is hard to glean any trend, linear or non-linear, from visualizing the raw data for most of the plots. Again, I defer to the editor to evaluate this judgment.

The authors truly appreciate the feedback from reviewer #2. The comments were instrumental in improving our manuscript.

Reviewer #3 (Remarks to the Author):

In general, exploring how epistasis changes with environmental conditions is a worthy goal. I appreciate the additional space given to the important idiosyncrasies of this particular system in the second revision (Lines 50-62). The paper (and responses to reviewers) are very clearly written, and since the revised paper itself does not exactly respond to several of the reviewer criticisms, I will trust that the authors are happy with it as it is. The work is quite professionally presented, and I'm sure it will be interesting to others working in epistatic theory.

The authors thank reviewer #3 for all their time, effort, and consideration.

To address the final concerns raised by Reviewer #3, we have expanded Box 1, which now summarizes our response to the reviewer regarding the role of higher-order interactions, as well as the potential effect of additional mutations on the magnitude of the weights ω_{ij} and the overall strength of global epistasis.